# Site-directed MT1-MMP trafficking and surface insertion regulate AChR clustering and remodeling at developing NMJs

Zora Chui-Kuen Chan[1], Hiu-Lam Rachel Kwan[1], Yin Shun Wong[2], Zhixin Jiang[1], Zhongjun Zhou[1], Kin Wai Tam[1], Ying-Shing Chan[1], Chi Bun Chan[2], Chi Wai Lee[1]*

[1]School of Biomedical Sciences, Li Ka Shing Faculty of Medicine, The University of Hong Kong, Hong Kong, China; [2]School of Biological Sciences, Faculty of Science, The University of Hong Kong, Hong Kong, China

**Abstract** At vertebrate neuromuscular junctions (NMJs), the synaptic basal lamina contains different extracellular matrix (ECM) proteins and synaptogenic factors that induce and maintain synaptic specializations. Here, we report that podosome-like structures (PLSs) induced by ubiquitous ECM proteins regulate the formation and remodeling of acetylcholine receptor (AChR) clusters via focal ECM degradation. Mechanistically, ECM degradation is mediated by PLS-directed trafficking and surface insertion of membrane-type 1 matrix metalloproteinase (MT1-MMP) to AChR clusters through microtubule-capturing mechanisms. Upon synaptic induction, MT1-MMP plays a crucial role in the recruitment of aneural AChR clusters for the assembly of postsynaptic specializations. Lastly, the structural defects of NMJs in embryonic MT1-MMP$^{-/-}$ mice further demonstrate the physiological role of MT1-MMP in normal NMJ development. Collectively, this study suggests that postsynaptic MT1-MMP serves as a molecular switch to synaptogenesis by modulating local ECM environment for the deposition of synaptogenic signals that regulate postsynaptic differentiation at developing NMJs.

**\*For correspondence:**
chiwai.lee@hku.hk

**Competing interests:** The authors declare that no competing interests exist.

## Introduction

Cell-cell communication in the nervous system occurs at specialized structures called synapses, where nerve signals propagate from a presynaptic neuron to a postsynaptic target cell. The postsynaptic apparatus containing a high density of neurotransmitter receptors represents an important feature of chemical synapses. In the past decades, tremendous effort has been made in understanding how the synthesis, trafficking, localization, and clustering of neurotransmitter receptors at the postsynaptic sites regulate synaptic formation, function, and plasticity in both central and peripheral synapses (*Li et al., 2018*; *Sanes and Lichtman, 2001*; *Song and Huganir, 2002*; *Wu et al., 2010*). Many of those studies were performed using the neuromuscular junction (NMJ), a simple peripheral chemical synapse. At the postsynaptic apparatus of NMJs, acetylcholine receptor (AChR) molecules are clustered at a nearly crystalline density of ~10,000 receptors/$\mu m^2$ (*Fertuck and Salpeter, 1976*); in contrast, its density drops drastically to <10 receptors/$\mu m^2$ at the extra-synaptic regions. However, the detailed molecular mechanisms underlying the formation and maintenance of densely clustered AChR molecules at the postsynaptic apparatus are not fully understood.

Previous knockout studies showed that even in the absence of motor neurons, AChR clusters can still be found in muscle fibers in vivo (*Lin et al., 2001*; *Yang et al., 2001*), suggesting that such AChR pre-patterns are spontaneously formed via muscle-intrinsic mechanisms. Interestingly, a similar structure of aneural AChR clusters can be induced in *Xenopus* primary muscles or immortalized C2C12 myotubes by culturing them on substratum coated with laminin, a major structural glycoprotein in the extracellular matrix (ECM) (*Kummer et al., 2004*; *Lee et al., 2009*). These aneural AChR

**eLife digest** Voluntary movement relies on skeletal muscle cells and nerve cells being able to communicate with one another. This communication occurs at a specialized region called the neuromuscular junction, or NMJ for short. These junctions are surrounded by a meshwork of proteins, known as the matrix, which structurally supports the nerve and muscle cells.

Muscle cells contain proteins called acetylcholine receptors on their cell surface. When these receptors cluster together at the NMJ, this allows nerve cells to communicate with the muscle cell and tell the muscle to contract. However, these clusters can also form spontaneously without the help of nerve cells at regions away from the communication site. Alongside these spontaneous clusters of acetylcholine receptors are dynamic actin-enriched structures. These structures are responsible for releasing enzymes that digest the surrounding matrix and are commonly found in migrating cells. But as skeletal muscle cells do not migrate, it remained unclear what purpose these structures serve at the NMJ.

Now, Chan et al. have used advanced microscopy techniques to show how these actin-enriched structures can help acetylcholine receptors cluster together at the site of communication between the nerve and muscle cells. The experiments showed that these structures direct a molecule called MT1-MMP to the muscle surface. This molecule then clears the surrounding matrix so that signals sent from the nerve can be effectively deposited at the narrow space between these two cells. When the muscle cells receive this initiating signal, acetylcholine receptors are recruited from the spontaneously formed clusters to the communication site, allowing the muscle to contract. When MT1-MMP was experimentally eliminated in mice, this disrupted the recruitment of acetylcholine receptors to the NMJ.

Overall, these experiments help researchers understand how clearing the matrix between nerve and muscle cells contributes to the deposition of factors that build the communication site at developing NMJs. In the future this might help develop treatments for movement disorders caused by abnormalities that affect the clearing of matrix proteins in these junctions.

clusters, located at the bottom surface of cultured muscles in direct contact with ECM proteins, may undergo topological transformation, which is mirrored by the progressive structural changes in synaptic AChR clusters during NMJ maturation in vivo (*Kummer et al., 2004*). Our previous work showed that actin-rich structures are highly concentrated at the perforated regions of aneural AChR clusters (*Lee et al., 2009*). These structures were later shown to share the typical characteristics of podosome-like structures (PLSs) (*Proszynski et al., 2009*). PLSs, initially identified as dynamic foot-like structures in motile or invasive cells, have been linked to pathophysiological processes such as cancer cell invasion and metastasis via local proteolysis of ECM proteins (*Linder, 2007*). It is worth to note that a small proportion of aneural AChR clusters can also be identified at the top surface of cultured muscles, and these spontaneously formed clusters are likely mediated through ECM- and PLS-independent mechanisms. At the NMJ, the exact functions of synaptic PLSs in regulating AChR cluster formation and remodeling remain largely unclear. In this study, we show that the assembly of PLSs, which can be induced by different ECM proteins, focally regulates matrix degradation for AChR clustering and topological remodeling. Next, we further demonstrate that intracellular trafficking and surface insertion of membrane-type 1 (MT1-) matrix metalloproteinase (MMP) are mediated via microtubule-capturing mechanisms at PLSs, which in turn spatiotemporally regulate the topological remodeling of aneural AChR clusters and their dispersal upon synaptic induction. Inhibition of MMP activity or reduced expression of muscle MT1-MMP greatly suppresses nerve-induced AChR cluster formation and stabilizes aneural AChR clusters against dispersal. Lastly, we show that MT1-MMP is required for the recruitment of AChR molecules from aneural to synaptic AChR clusters at developing NMJs in vitro and in vivo. Taken together, this study revealed the significance of PLS-directed MT1-MMP trafficking and surface insertion in modulating the assembly and topological remodeling of AChR clusters via focal matrix degradation at developing neuromuscular synapses.

# Results

## Topologically complex structures of PLS-associated aneural AChR clusters can be induced by different ECM proteins

When dissociated myotomal tissues from early *Xenopus* embryos were cultured on glass coverslips coated with a mixture of ECM proteins, containing entactin/nidogen, collagen and laminin (ECL), topologically complex aneural AChR clusters were observed mostly on the bottom surface of muscle cells in contact with ECL-coated substratum (*Figure 1A*). Some aneural AChR clusters could also be found within the top surface of muscle cells but exhibited a sparsely scattered morphology. To further identify if specific ECM proteins are required for the formation of these topologically complex

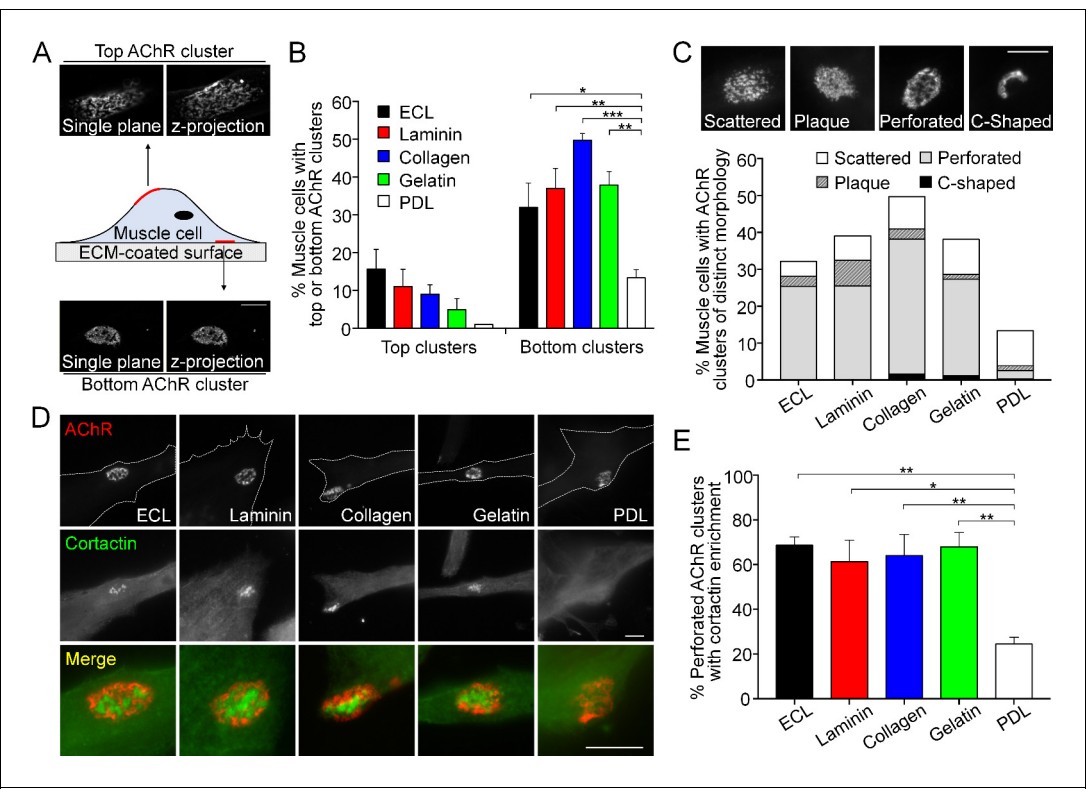

**Figure 1.** Topologically complex structures of PLS-associated aneural AChR clusters can be induced by different ECM proteins. (**A**) A schematic diagram illustrating the location and morphological features of aneural AChR clusters found in the top versus bottom surface of cultured *Xenopus* muscle cells. Maximal projection of confocal z-stack images was constructed from 11 frames with 3 μm z-interval. (**B**) Quantification on the presence of top versus bottom aneural AChR clusters in muscle cells cultured on substratum coated with different ECM proteins or PDL. n = 300 cells in each condition from 3 independent experiments. (**C**) Representative images showing several distinct morphological features of aneural AChR clusters (scattered, plaque, perforated, and C-shaped), and the quantification showing the population of aneural AChR clusters with respective morphological features in muscle cells cultured on different ECM substrates or PDL. n = 150 cells in each condition from 3 independent experiments. (**D**) Representative images showing cortactin immunostaining of 2-d old *Xenopus* muscle cells to indicate the presence of PLSs at the perforated regions of aneural AChR clusters. (**E**) Quantification on the percentage of aneural AChR clusters associated with cortactin enrichment in muscle cells cultured on substratum coated with different ECM proteins or PDL. n = 150 cells in each condition from 3 independent experiments. Scale bars represent 10 μm. Data are represented as mean ± SEM. One-way ANOVA with Dunnett's multiple comparisons test, *, **, *** represent p≤0.05, 0.01, and 0.001 respectively.

The online version of this article includes the following figure supplement(s) for figure 1:

**Figure supplement 1.** Most aneural AChR clusters are gradually dispersed in cultured *Xenopus* muscle cells over time.

**Figure supplement 2.** PLS core and cortex markers are present at ECM-induced AChR clusters in *Xenopus* primary muscle cultures.

AChR clusters, we tested several key ECM proteins individually, including laminin, collagen, and gelatin, for their ability to induce AChR cluster formation in cultured *Xenopus* muscle cells. We found that the formation of bottom AChR clusters could be effectively induced by all ECM proteins tested, in contrast to the negative control using poly-D-lysine (PDL), a polypeptide commonly used to promote cell attachment (*Figure 1B*). In immortalized C2C12 myotubes, aneural AChR clusters can undergo a topological transformation from a plaque to a perforated pretzel-shaped, and eventually to C-shaped arrays (*Kummer et al., 2004*). Hence, we further classified the aneural AChR clusters in *Xenopus* muscle cultures into scattered, plaque, perforated, and C-shaped based on their morphological features. A majority of ECM-induced aneural AChR clusters exhibited perforated structures. While the percentage of muscle cells with AChR clusters was largely reduced on PDL-coated substrate, the largest share of AChR clusters exhibited scattered structures (9.59% out of 13.33% total) (*Figure 1C*), similar to those structures found on the top muscle surface. As opposed to a previous study showing the topological transformation of AChR clusters in C2C12 myotubes (*Kummer et al., 2004*), most AChR clusters, regardless of their topological features, were gradually dispersed over 6 days in cultured *Xenopus* muscle cells (*Figure 1—figure supplement 1*).

Our previous study indicated that actin depolymerizing factor (ADF)/cofilin, one of the molecular components in PLSs, regulates the formation and maintenance of AChR clusters via vesicular trafficking (*Lee et al., 2009*). Here, we further performed immunostaining of several typical core markers (composed of F-actin and actin-associated proteins, such as ADF/cofilin, Arp2/3 complex, and cortactin) and cortex markers (composed of adhesion molecules, such as vinculin, talin, and paxillin) of PLSs. All these markers were found to be enriched in the perforations (arrows) of AChR clusters in *Xenopus* muscle cultures (*Figure 1—figure supplement 2*). As cortactin is highly enriched and involved in both early and late stages of PLS formation (*Webb et al., 2006*), we then used cortactin as an indicator of PLS localization in AChR clusters. PLSs were spatially concentrated in a majority of perforated AChR clusters induced by different ECM proteins (*Figure 1D and E*). In contrast, PLSs were not localized in AChR clusters, even in a small fraction of AChR clusters (2.16% out of 13.33% total) exhibiting perforated structures, on PDL substrates. These data suggest that different ECM proteins can induce the assembly of PLSs, which may in turn promote the formation of topologically complex structures of AChR clusters.

## MMP-mediated ECM degradation regulates topological remodeling of aneural AChR clusters

The data above showed that perforated AChR clusters can be effectively induced by gelatin (*Figure 1B and C*), an ECM protein that was previously shown to induce simple AChR plaques only (*Kummer et al., 2004*). This novel finding prompted us to adopt the well-established fluorescent gelatin degradation assay (*Starnes et al., 2011*) to investigate the possible association between matrix degradation and PLS localization in AChR clusters. After plating the muscle cells on FITC-gelatin-coated substratum, confocal microscopy was performed to visualize the extent of gelatin degradation in 3 day old live *Xenopus* primary muscle cells. We observed FITC-gelatin that was extensively degraded in close association with the perforated regions of aneural AChR clusters (*Figure 2A*). Next, we quantitatively measured the fluorescence intensity of FITC-gelatin in different AChR clusters and identified a linear relationship between the extent of gelatin degradation and the perforation area of aneural AChR clusters (*Figure 2B*). A similar spatially restricted gelatin degradation pattern was observed in cultured C2C12 myotubes (*Figure 2C*). Besides gelatin, immunostaining experiments also showed the spatial degradation of laminin at the perforated regions of AChR clusters in *Xenopus* primary muscle cells (*Figure 2D*).

To examine if the focal degradation pattern of ECM proteins is mediated by MMP activity, we first used two broad-spectrum MMP inhibitors, BB-94 and Marimastat (BB-2516), which are known to be highly potent against MMP-1, -2, -3, -7, -9 and MT1-MMP (*Cathcart et al., 2015*). Although perforated AChR clusters could be identified in muscle cells treated with 5 µM BB-94 or 10 µM BB-2516, the degradation of fluorescent gelatin associated with these perforated AChR clusters was completely abolished (*Figure 2E and F*). Quantification further demonstrated that the initial formation of aneural AChR clusters in muscle cells was largely unaffected in the presence of MMP inhibitors (*Figure 2G*). To investigate if PLS-associated MMP activity regulates the topological remodeling of AChR clusters, we monitored the same group of muscle cells over 2 days after aneural AChR clusters had been formed. We observed a significant reduction in the number of aneural AChR clusters

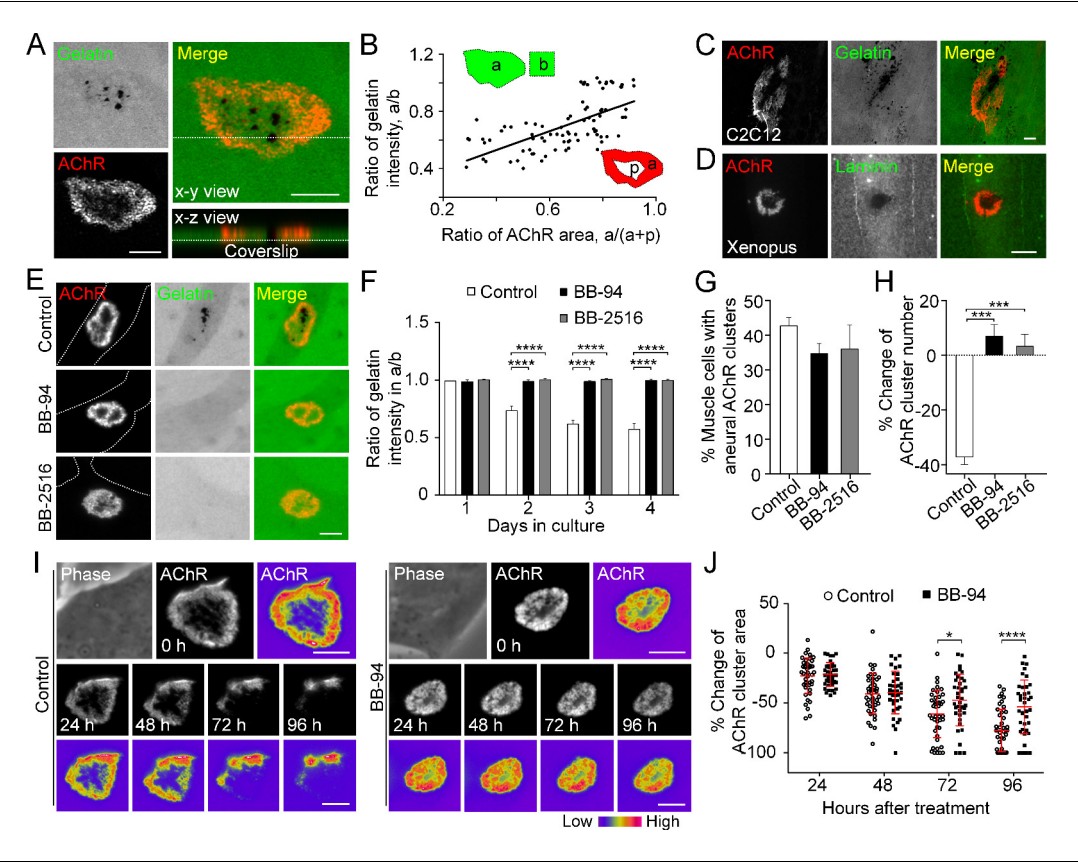

**Figure 2.** MMP-mediated ECM degradation regulates topological remodeling of aneural AChR clusters. (**A**) Representative confocal z-stack images showing the discrete and extensive degradation of fluorescent gelatin at the perforated regions within aneural AChR clusters. (**B**) Scatter plot analysis showing a positive correlation between the fluorescent gelatin intensity in perforated AChR clusters (a) divided by that in another region of the same cell (b) and the area of AChR-rich region (a) divided by that of the entire AChR clusters with perforations (a +p). R = 0.6458, p<0.0001. n = 84 from 3 independent experiments. (**C–D**) Representative images showing the degradation patterns of fluorescent gelatin (**C**) and laminin immunostaining (**D**) at AChR clusters in cultured C2C12 myotubes and *Xenopus* primary muscle cells, respectively. (**E**) Representative images showing the inhibitory effects of BB-94 or BB-2516 on fluorescent gelatin degradation. Dotted lines indicate the periphery of muscle cells. (**F**) Quantification on the extent of fluorescent gelatin degradation at the perforated regions of AChR clusters in control and BB-94/BB-2516-treated muscle cells over the first 4 consecutive days in culture. n > 280 muscle cells in each condition from 3 independent experiments. (**G**) Quantification on the effects of BB-94 or BB-2516 on the formation of bottom AChR clusters in cultured *Xenopus* muscle cells. n = 150 muscle cells in each condition from 3 independent experiments. (**H**) Quantification on the effects of BB-94 or BB-2516 on the stability of bottom AChR clusters in cultured *Xenopus* muscle cells. n = 166 (Control), 154 (BB-94), and 137 (BB-2516) muscle cells from 3 independent experiments. (**I**) Representative sets of time-lapse images showing the topological changes and fluorescence intensity of AChR clusters in control (left panels) and BB-94-treated (right panels) muscle cells. 8-bit pseudo-color images highlight the difference in AChR intensity at the same aneural clusters over 96 hr with or without BB-94 treatment. (**J**) An individual value plot showing the percentage change in the area of each aneural AChR clusters at different time-points between control and BB-94-treated cells. n = 43 (Control) and 38 (BB-94) muscle cells from 3 independent experiments. Data are represented as mean ± SD. Scale bars represent 5 μm. Data are represented as mean ± SEM, unless otherwise specified. Two-way ANOVA with Dunnett's multiple comparisons test (**F**), one-way ANOVA with Dunnett's multiple comparisons test (**G and H**), two-way ANOVA with Sidak's multiple comparisons test (**J**), *, ***, **** represent p≤0.05, 0.001, and 0.0001 respectively.

in control muscle cells, in contrast to BB-94 or BB-2516 treatment that caused a slight increase in the number of aneural AChR clusters (*Figure 2H*). To further explore if MMP activity regulates the topological remodeling of AChR clusters, we performed time-lapse imaging to monitor the dynamic changes in AChR cluster morphology and intensity in response to BB-94 treatment. AChR clusters

were first labeled with fluorescent α-bungarotoxin (0 hr), then they were monitored for 96 hr after treatment. In control untreated cells, we observed a gradual dispersal of aneural AChR clusters over the 96 hr imaging period; however, such spontaneous dispersal and topological remodeling of AChR clusters were largely inhibited by BB-94 treatment within the same imaging period (*Figure 2I and J*). These data demonstrated that MMP activity is involved in the topological remodeling and dispersal of aneural AChR clusters.

## MT1-MMP activity precisely controls the extent of ECM degradation and AChR cluster formation

The spatially restricted patterns of gelatin degradation suggested the possible involvement of membrane-type MMPs (MT-MMPs) in modulating ECM environment focally to regulate AChR cluster remodeling and dispersal. To test this, we first performed immunostaining of *Xenopus* muscle cells and detected endogenous MT1-MMP that was enriched in the perforations of aneural AChR clusters (*Figure 3A*). Next, we over-expressed MT1-MMP-mCherry construct in muscle cells via *Xenopus* embryo microinjection. In MT1-MMP-mCherry-overexpressing muscle cells, we observed extensive degradation of fluorescent gelatin in the entire cell area (*Figure 3B*), in contrast to the spatially restricted degradation pattern at the perforated regions of AChR clusters in control cells. In addition, MT1-MMP-mCherry overexpressing cells showed a lower percentage of ECM-induced bottom AChR clusters, but a higher percentage of top AChR clusters, in comparison to the wild-type control cells (*Figure 3C*). Quantitative analyses on those bottom AChR clusters in MT1-MMP-mCherry-overexpressing cells showed that the intensity of aneural AChR clusters was significantly reduced, which was accompanied by the extensive gelatin degradation (*Figure 3D*). Interestingly, BB-94 or BB-2516 treatment was able to partially revert the distribution of aneural AChR clusters from the top surface back to the bottom surface of MT1-MMP-mCherry-overexpressing muscle cells (*Figure 3C*). Consistent with that, the extensive degradation of fluorescent gelatin and the reduction of AChR intensity caused by MT1-MMP-mCherry overexpression were largely rescued by either BB-94 or BB-2516 treatment (*Figure 3D*). Our results indicated that the precise control of focal ECM degradation by MT1-MMP at the perforated sites plays an essential role in the formation of bottom aneural AChR clusters.

## Cortical microtubule capturing is mediated by EB1-/CLASP-dependent mechanisms at perforated AChR clusters

To investigate the cytoskeletal involvement for intracellular trafficking of MT1-MMP, we next performed total internal reflection fluorescence (TIRF) microscopy to visualize the perimembrane fraction of GFP-tagged microtubule plus end-binding protein 1 (EB1-GFP) in live cultured muscle cells. Interestingly, we observed EB1-GFP comets to be highly enriched at the perforated aneural AChR clusters (*Figure 4A*). This pattern of EB1-GFP comets was not an artifact of overexpression, as a similar localization pattern of endogenous EB1 signals was also detected in fixed *Xenopus* muscle cells (*Figure 4—figure supplement 1*). By analyzing the density and trajectory of EB1-GFP comets in time-lapse images, we found that EB1-GFP comets were densely localized, but they showed lower mobility, at sites of aneural AChR clusters than other regions of the cell (*Figure 4B and C*). These data suggested that AChR clusters are the focal point of directed microtubule-based intracellular transport. Previous studies showed that cytoplasmic linker-associated protein (CLASP) and LL5β, a PLS cortex protein involved in NMJ development (*Kishi et al., 2005*), interact with microtubules to direct the vesicular trafficking of different proteins to the NMJ (*Basu et al., 2015*). Thus, we examined if CLASP is required for directing the movement of EB1-GFP comets to PLS-enriched AChR clusters using antisense morpholino oligonucleotide (MO)-mediated knockdown approach. As validated by western blot analysis, endogenous CLASP protein level was largely reduced in *Xenopus* embryos microinjected with CLASP-MO (*Figure 4—figure supplement 2*). In CLASP-MO muscle cells, TIRF microscopy showed the perimembrane signals of EB1-GFP that were primarily found at the center of perforations within aneural AChR clusters (*Figure 4A*). Although the density of EB1-GFP comets was only slightly reduced at AChR cluster region (*Figure 4B*), the average speed of EB1-GFP comets in the region of AChR clusters was significantly increased by 26.7% to 0.12 ± 0.004 μm/s in CLASP-MO muscle cells (*Figure 4C*), indicating that CLASP is involved in microtubule capturing at aneural AChR clusters.

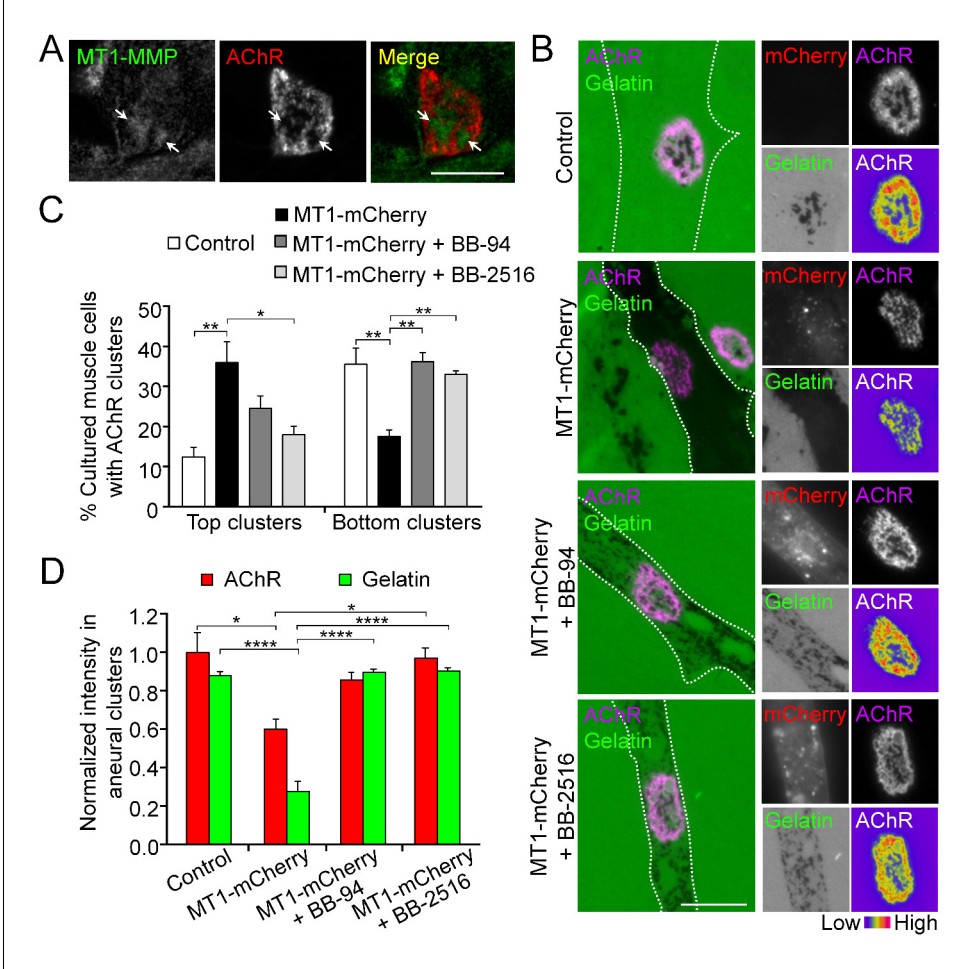

**Figure 3.** MT1-MMP activity precisely controls the extent of ECM degradation and AChR cluster formation. (**A**) Representative images showing the spatial localization of endogenous MT1-MMP in perforations of aneural AChR clusters (arrows) in fixed *Xenopus* muscle cultures. (**B**) Representative images showing the extent of fluorescent gelatin degradation in area covered by MT1-MMP-mCherry-overexpressing (MT1-mCherry) muscle cells in the presence or absence of MMP inhibitors BB-94 or BB-2516. 8-bit pseudo-color images highlight the relative fluorescence intensity of AChR clusters in different conditions. (**C**) Quantification on the effects of MT1-MMP-mCherry overexpression on the formation of AChR top and bottom clusters in response to BB-94 or BB-2516 treatment. n = 163 (Control), 123 (MT1-mCherry), 119 (MT1-mCherry + BB-94), and 62 (MT1-mCherry + BB-2516) muscle cells from 4 independent experiments. (**D**) Quantification on the effects of MT1-MMP-mCherry overexpression on the extent of fluorescent gelatin degradation and the intensity of aneural AChR clusters in response to BB-94 or BB-2516 treatment. For gelatin intensity measurement: n = 30 (Control), 39 (MT1-mCherry), 33 (MT1-mCherry + BB-94), and 31 (MT1-mCherry + BB-2516) muscle cells from 3 independent experiments. For AChR intensity measurement: n = 14 (Control), 19 (MT1-mCherry), 21 (MT1-mCherry + BB-94), and 16 (MT1-mCherry + BB-2516) muscle cells from 3 independent experiments. Scale bars represent 10 μm. Data are represented as mean ± SEM. One-way ANOVA with Dunnett's multiple comparisons test (**C**), one-way ANOVA with Turkey's multiple comparisons test (**D**), *, **, **** represent p≤0.05, 0.01, and 0.0001 respectively.

To further study the directed trafficking and local capturing of EB1-GFP at aneural AChR clusters, we performed TIRF in combination with fluorescence recovery after photobleaching (FRAP) experiments. Before photobleaching, EB1-GFP signals were primarily enriched at the edge of perforations of aneural AChR clusters in control cells but at the center of perforations in CLASP-MO muscle cells (*Figure 4D*). To better show the spatial patterns of EB1-GFP signals at the perforations of aneural AChR clusters in control versus CLASP-MO muscle cells, we have plotted the fluorescence intensity profiles of EB1-GFP and AChR across a perforated region of AChR clusters, as indicated in *Figure 4D*. In the control cell, the peaks of EB1-GFP intensity showed a slight lateral shift towards

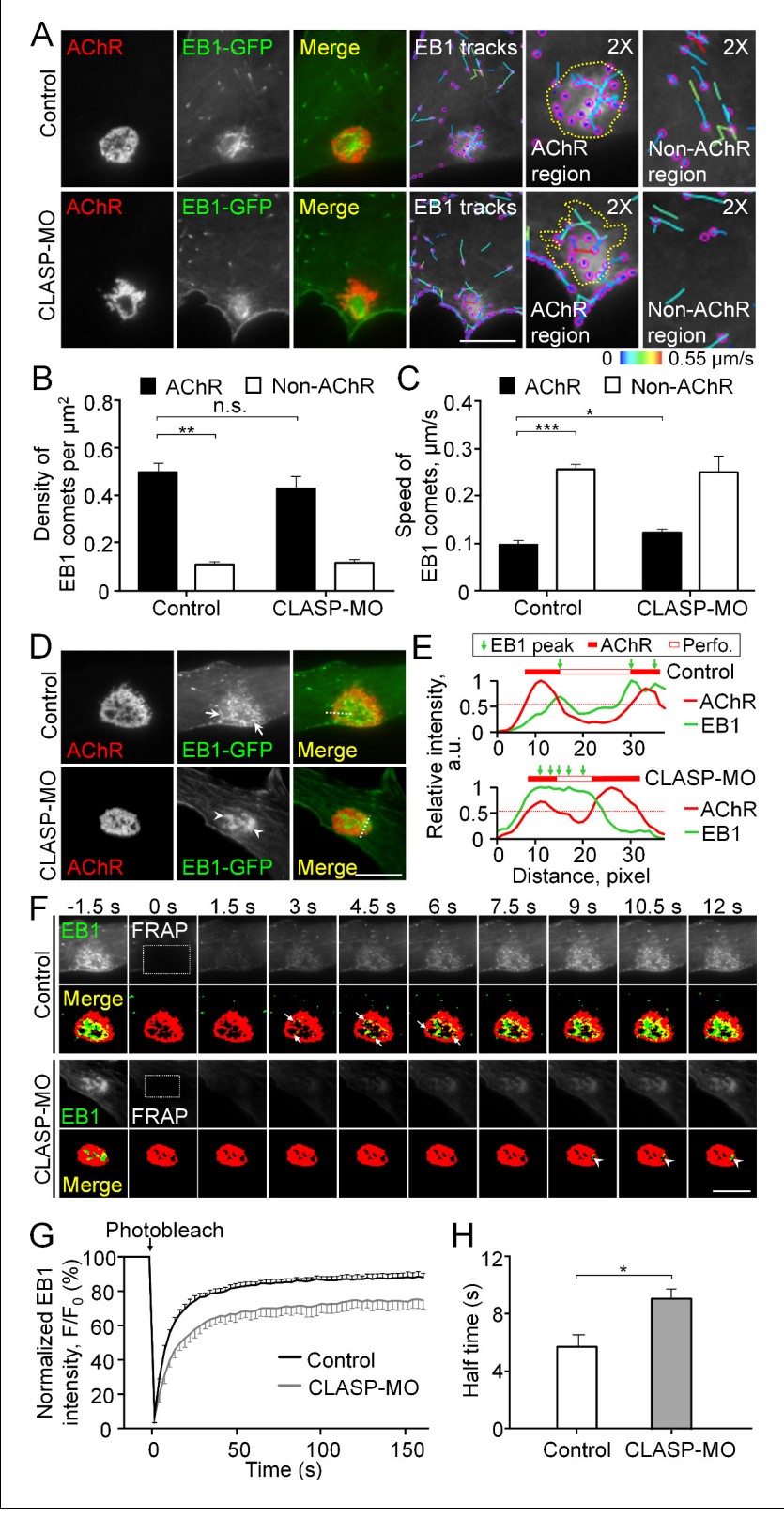

**Figure 4.** Cortical microtubule capturing is mediated by EB1-/CLASP-dependent mechanisms at perforated AChR clusters. (**A**) Representative TIRF images showing the density and speed of EB1-GFP comets in AChR cluster versus and non-AChR regions between control and CLASP-MO muscle cells. The trajectories of EB1-GFP comets (purple dots) were constructed from 7 frames in 18 s, and then color-coded to reflect their speed. Yellow dotted lines

*Figure 4 continued on next page*

*Figure 4 continued*

indicate the location of AChR clusters. (**B–C**) Quantitative analyses showing the density (**B**) and speed (**C**) of EB1-GFP comets in AChR cluster and non-AChR regions between control and CLASP-MO muscle cells. n = 16 (Control), and 13 (CLASP-MO) cells from 3 independent experiments. (**D**) Representative TIRF images showing the differential perimembrane localization of EB1-GFP signals at aneural AChR clusters in control and CLASP-MO muscle cells. Arrows and arrowheads indicate the spatial enrichment of EB1-GFP signals at AChR clusters. (**E**) Line profiles showing the relative fluorescence intensities of AChR (red) and EB1-GFP (green) signals along the dotted lines indicated in merge image in panel D. The perforated region was defined by a cutoff intensity (55%) of the maximum, and the EB1 peaks were marked based on the line profiles. (**F**) Representative sets of time-lapse images showing the fluorescence recovery of EB1-GFP signals after photobleaching the region of aneural AChR clusters (dotted rectangle) in control and CLASP-MO muscle cells. Arrows and arrowheads indicate the spatial recovery of EB1-GFP signals in control and CLASP-MO muscle cells, respectively. (**G–H**) Quantification showing the FRAP curves (**G**) and the calculated recovery half time (**H**) of EB1-GFP signals in control versus CLASP-MO muscle cells. n = 18 (Control) and 14 (CLASP-MO) cells from 4 independent experiments. Scale bars represent 10 μm. Data are represented as mean ± SEM. Student's t-test, \*, \*\*, \*\*\* represent p≤0.05, 0.01, and 0.001 respectively. n. s.: non-significant.

The online version of this article includes the following video and figure supplement(s) for figure 4:

**Figure supplement 1.** Immunostaining demonstrates the localization of endogenous EB1 in cultured *Xenopus* muscle cells.

**Figure supplement 2.** Endogenous CLASP protein level is effectively reduced by specific antisense morpholino oligonucleotides.

**Figure 4—video 1.** Time-lapse video showing the FRAP signals of EB1-GFP at the perforated aneural AChR clusters in a control muscle cell.

https://elifesciences.org/articles/54379#fig4video1

**Figure 4—video 2.** Time-lapse video showing the FRAP signals of EB1-GFP at the perforated aneural AChR clusters in a CLASP-MO muscle cell.

https://elifesciences.org/articles/54379#fig4video2

the center of a perforated region (Top chart, *Figure 4E*), indicating the preferential enrichment of EB1-GFP at the edge of perforations. In contrast, CLASP knockdown caused the peak of EB1-GFP intensity to cover a larger area of a perforated region, indicating that EB1-GFP is enriched relatively closer to the center of perforations (Bottom chart, *Figure 4E*). After photobleaching the region of perforated AChR clusters (dotted rectangles in *Figure 4F*), we performed time-lapse imaging that reflects the rapid recovery of EB1-GFP fluorescence signals (in seconds after photobleaching). Spatially, the recovery of EB1-GFP signals was detected near the edge of those perforations within AChR clusters in control cells (arrows in *Figure 4F*, *Figure 4—video 1*), but it was largely reduced in CLASP-MO muscle cells (arrowheads in *Figure 4F*, *Figure 4—video 2*), leading to a significant increase in the half-time of EB1-GFP signal recovery by 58% from 5.71 ± 0.83 s in control cells to 9.01 ± 0.71 s in CLASP-MO muscle cells (*Figure 4G and H*). Collectively, our data suggested that PLSs may direct the vesicular trafficking to AChR clusters through EB1-/CLASP-mediated microtubule-capturing mechanisms.

## Intracellular trafficking of MT1-MMP is directed to PLS-associated AChR clusters

As microtubules serve as major tracks for vesicular trafficking in mammalian cells, the cortical microtubule organization plays a crucial role in the targeted delivery of secretory and membrane proteins via exocytosis (*Noordstra and Akhmanova, 2017*). To further investigate if MT1-MMP intracellular trafficking is mediated by microtubule-based transport, we performed dual-channel live-cell imaging on *Xenopus* muscle cells over-expressing both MT1-MMP-mCherry and EB1-GFP (*Figure 5A*). In region 'i', some MT1-MMP-mCherry vesicles (arrows) were found to be initially immobile at the beginning of this time-lapse series, until they were captured by a moving EB1-GFP comet (arrowhead), leading to a coordinated movement of both EB1-GFP and MT1-MMP-mCherry signals from 27 s to 29 s time-points in this imaging period (middle row, *Figure 5A*). Another example in region 'ii' indicated that MT1-MMP-mCherry vesicles (arrows) were able to move bidirectionally on microtubule structures, as visualized by the fading tracks (arrowheads) generated by moving EB1-GFP comets (bottom row, *Figure 5A*). The spatiotemporal correlation between MT1-MMP-mCherry and EB1-

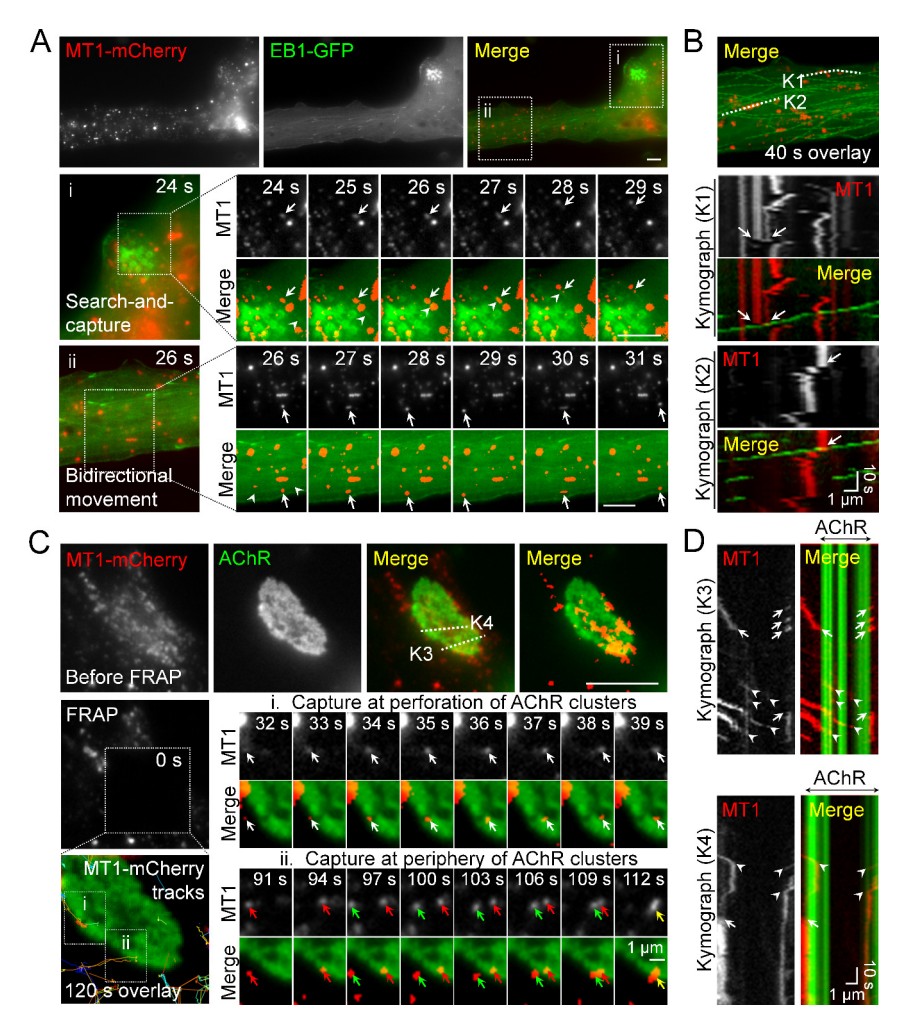

**Figure 5.** Intracellular trafficking of MT1-MMP is directed to PLS-associated AChR clusters. (**A**) Representative images showing the intracellular trafficking of MT1-MMP vesicles in a cultured muscle cell expressing both MT1-MMP-mCherry (MT1-mCherry) and EB1-GFP. The time-lapse series of two regions (i and ii) outlined in the merge image showed (i) the search-and-capture of MT1-mCherry vesicles (arrows) by an EB1-GFP comet (arrowheads); and (ii) the bidirectional movement of MT1-mCherry vesicles (arrows) along the microtubules (arrowheads). (**B**) Kymographs showing the spatiotemporal correlation between MT1-mCherry and EB1-GFP signals along K1 and K2 lines indicated in the merge image. The maximal projection of MT1-mCherry and EB1-GFP signals was constructed from 40 frames in a 40 s time-lapse series. Arrows indicate the examples of lateral displacement of initially stationary MT1-mCherry vesicles after EB1-GFP comets had passed through. (**C**) Representative TIRF-FRAP images showing the local capturing of MT1-mCherry vesicles at AChR clusters. After photobleaching, the recovery of MT1-mCherry signals and their trajectories in two regions of interest indicate (i) MT1-mCherry vesicles (arrows) were transported to and captured at the perforation of AChR clusters; and (ii) two groups of MT1-mCherry vesicles (red and green arrows) were transported to and captured at the same site of AChR cluster periphery over a period of 120 s. (**D**) Kymographs showing the spatiotemporal capture of MT1-mCherry vesicles at AChR clusters. Two kymographs were constructed from 120 time-lapse images along K3 and K4 lines, as indicated in the merge image (top panels in C). Arrowheads and arrows indicate the sites of MT1-mCherry capture at the perforated and peripheral regions of AChR clusters, respectively. Scale bars represent 10 μm, unless stated otherwise.

The online version of this article includes the following figure supplement(s) for figure 5:

**Figure supplement 1.** Immunostaining demonstrates the vesicular localization of endogenous MT1-MMP in cultured *Xenopus* muscle cells.

GFP signals was better appreciated by constructing kymographs using multiple timeframes along 'K1' and 'K2' lines (*Figure 5B*). In these two kymographs, most MT1-MMP-mCherry vesicles were found to be relatively immobile. Interestingly, we observed some displacement of MT1-MMP-mCherry vesicles immediately after EB1-GFP comets had come into contact with them (arrows in *Figure 5B*). Consistent with the subcellular localization of MT1-MMP-mCherry, immunostaining data revealed that endogenous MT1-MMP exhibited punctuated structures inside the muscle cells, and some of them were co-localized with vesicle-associated membrane protein 1 (VAMP1) (*Figure 5—figure supplement 1*). Taken together, these data illustrated that EB1 coordinates the vesicular trafficking of MT1-MMP in cultured muscle cells.

To further visualize the directed trafficking and capturing of MT1-MMP vesicles at PLS-associated AChR clusters, we performed TIRF-FRAP experiments on MT1-MMP-mCherry-expressing muscle cells after AChR labeling. The cells with low expression level of MT1-MMP-mCherry were chosen, in which the formation of aneural AChR clusters was largely unaffected. After photobleaching, we detected local capturing of MT1-MMP-mCherry vesicles at the edge of perforations within aneural AChR clusters (white arrows, region 'i' in *Figure 5C*), as well as at the edge of AChR cluster periphery (red and green arrows, region 'ii' in *Figure 5C*). In kymographs constructed along 'K3' and 'K4' lines, we observed MT1-MMP-mCherry vesicles that were frequently captured at both perforations (arrowheads) and periphery (arrows) of AChR clusters over the 120 s imaging period after photobleaching (*Figure 5D*). As both sites are enriched with PLS cortex markers, vinculin and talin (arrowheads in *Figure 1—figure supplement 2*), we speculated that PLSs, through the cortex domains, can mediate site-directed capturing of MT1-MMP vesicles for their subsequent surface insertion.

## Surface insertion of MT1-MMP is correlated with the formation and remodeling of AChR clusters

As MT1-MMP-mCherry over-expression significantly inhibited the formation of AChR bottom clusters (*Figure 3C*), we further used MT1-MMP tagged with a pH-sensitive green fluorescent protein pHluorin (MT1-MMP-pHluorin) to verify if the surface expression of MT1-MMP produces a similar inhibitory effect on AChR clustering. In cultured muscle with low MT1-MMP-pHluorin expression level, MT1-MMP-pHluorin signals were spatially enriched at the perforated regions of aneural AChR clusters (arrow ) and caused only a slight inhibition on aneural AChR clustering (*Figure 6A and B*). At high MT1-MMP-pHluorin expression, the formation of AChR bottom clusters was significantly inhibited, leading to sparsely scattered appearance if observed. These data further confirmed that MT1-MMP activity is crucial for the formation of topologically complex AChR clusters by precisely controlling focal ECM degradation at the perforated sites.

To validate the surface localization of MT1-MMP in AChR clusters, we treated MT1-MMP-pHluorin-expressing muscle cells with a non-permeable buffer at pH 5.0 and observed a large reduction in MT1-MMP-pHluorin fluorescence intensity (arrows in *Figure 6C*), mimicking the quenching of MT1-MMP-pHluorin signals in acidic vesicular compartments. The signals were then restored after changing the culture medium back to pH 7.8, validating the effectiveness of this probe for visualizing surface localization of MT1-MMP, as it emits bright fluorescence upon surface insertion. Using this probe, we further determined the correlation between surface localization of MT1-MMP and topological remodeling of AChR clusters by monitoring the changes in the same AChR clusters over 24 hr in cultured muscle cells with different expression levels of MT1-MMP-pHluorin (*Figure 6D*). It should be noted that muscle cells with low MT1-MMP-pHluorin expression level were chosen to minimize the non-specific effects of exogenous overexpression. In this example, AChR cluster intensity was moderately reduced by 26% after 24 hr in the muscle cell with a very low expression level of MT1-MMP-pHluorin. In contrast, AChR cluster intensity in a muscle cell with a relatively higher (low) MT1-MMP-pHluorin expression showed about 78% reduction during the same period. Quantitative analyses on a pool data of 13 AChR clusters further indicated that MT1-MMP-pHluorin intensity in AChR cluster region was linearly correlated with the change in AChR cluster intensity over the next 24 hr (*Figure 6E*). These data indicated the functional role of surface MT1-MMP in promoting the topological remodeling of AChR clusters.

Next, we performed TIRF-FRAP experiments that aimed to identify the exact locations of MT1-MMP surface insertion at AChR clusters. In this example, strong and discrete MT1-MMP-pHluorin signals re-appeared at the perforation (arrows) and periphery (arrowheads) of AChR clusters at time-points of 118 s and 48 s, respectively, in control muscle cells after photo-bleaching (*Figure 6F*).

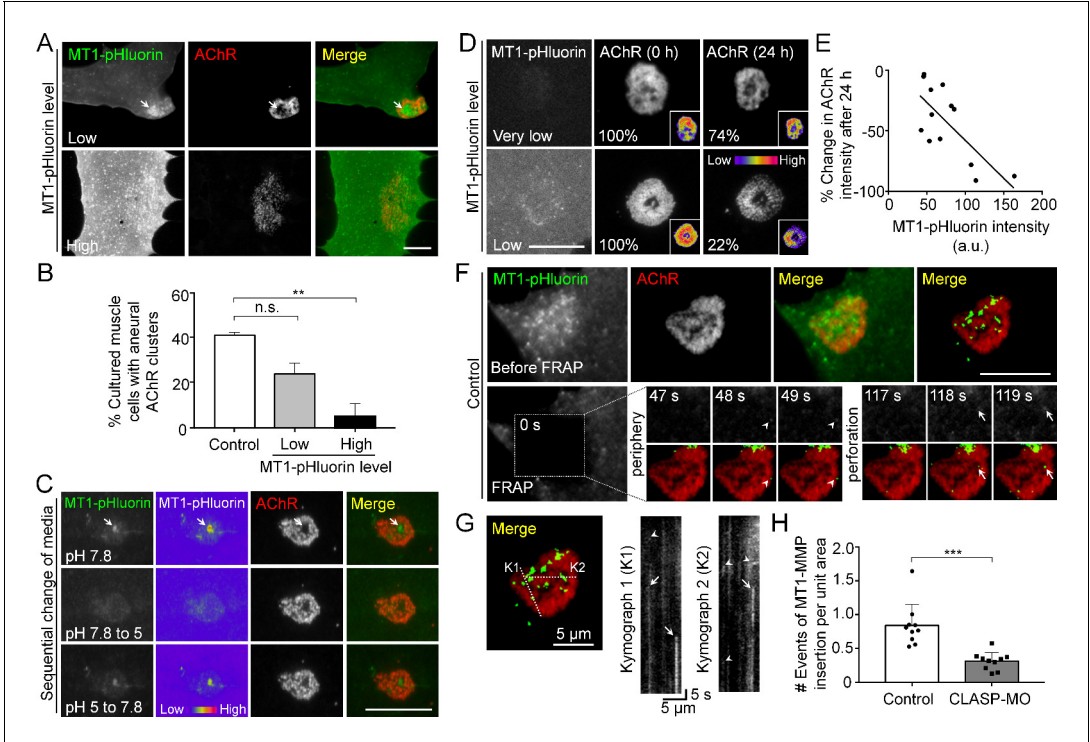

**Figure 6.** Spatial delivery of surface MT1-MMP regulates AChR cluster formation and remodeling. (**A**) Representative images showing the effects on aneural AChR clustering in cultured muscle cells over-expressing different levels of MT1-MMP-pHluorin (MT1-pHluorin). Arrows indicate the spatial localization of MT1-pHluroin at perforated regions of AChR clusters. (**B**) Quantification on the effects on aneural AChR cluster formation in cultured muscle cells with different levels of MT1-pHluorin over-expression. Cultured muscle cells with average MT1-pHluorin intensity above the cutoff value of >500 (arbitrary unit) were classified as high expression. n = 150 (Control), 32 (Low MT1-pHluorin level), and 15 (High MT1-pHluorin level) muscle cells from 3 independent experiments. (**C**) Representative sets of images showing the change in MT1-pHluorin fluorescence intensity in response to a sequential switching of various culture media with different pH levels. Arrows indicate the spatial localization of MT1-pHluroin signals in the perforated region of AChR clusters. 8-bit pseudo-color images highlight the change in the relative intensity of MT1-pHluorin. (**D**) Representative sets of time-lapse images showing the extent of AChR cluster remodeling in muscle cells with different MT1-pHluorin expression levels. 8-bit pseudo-color images in the insets highlight the change in the relative intensity of AChR clusters. The percentage values indicate the relative fluorescence intensity of AChR clusters after 24 hr. (**E**) A scatter plot analysis showing the correlation between MT1-pHluorin intensity and the percentage change in AChR intensity over 24 hr in cells with different MT1-pHluorin expression levels. The black line indicates a linear correlation. R = −0.7188, p=0.0056. n = 13 from 3 independent experiments. (**F**) Representative TIRF-FRAP images showing the surface insertion of MT1-pHluorin at AChR clusters in cultured muscle cells. Arrows and arrowheads indicate the spatial insertion of MT1-pHluorin at the perforation and periphery of AChR clusters, respectively. (**G**) Kymographs showing the spatiotemporal insertion of MT1-pHluorin at AChR clusters. Two kymographs were constructed from 180 time-lapse images along the lines of K1 and K2, as indicated in the merge image (left panel). Arrows indicate some newly inserted MT1-pHluorin that were relatively stable, while arrowheads indicate that some were dispersed shortly after surface insertion. (**H**) Quantification on the number of events of MT1-pHluorin surface insertion per unit area (μm²) of aneural AChR clusters over the entire time-lapse duration. n = 10 (Control) and 10 (CLASP-MO) muscle cells from 3 independent experiments. Scale bars represent 10 μm, unless otherwise specified. Data are represented as mean ± SEM. One-way ANOVA with Bonferroni's multiple comparisons test (B), Student's t-test (H), **, *** represent p≤0.01 and 0.001 respectively. n.s.: non-significant.

Kymographs constructed along 'K1' and 'K2' lines indicated that some newly inserted MT1-MMP-pHluorin signals were relatively stable (arrows), while some were short-lived but with multiple events of MT1-MMP surface insertion at the same location in AChR clusters (arrowheads) within a short imaging period (*Figure 6G*). By quantifying the number of events of MT1-MMP surface insertion, we detected a significant reduction of MT1-MMP-pHluorin surface insertion at AChR clusters in CLASP-MO muscle cells when compared to that in control cells (*Figure 6H*). Together with the above data showing the local capture of EB1-GFP comets, this data further confirmed that vesicular trafficking and surface targeting of MT1-MMP are spatially regulated by PLS cortex proteins at the perforation and periphery of AChR clusters through EB1-/CLASP-mediated microtubule capturing mechanisms.

# Pharmacological inhibition of MMP activity suppresses the assembly of synaptic AChR clusters and the disassembly of aneural AChR clusters

To understand the functional role of PLS-associated MMP activity at developing NMJs, we first examined the effects of MMP inhibitors on ECM degradation and synaptic AChR clustering in *Xenopus* nerve-muscle co-cultures. Along the trail of neurites in contact with the basal membrane of muscle cells (arrows), we observed spatial degradation of fluorescent gelatin-coated substratum and synaptic AChR clustering at the nerve-muscle contacts (*Figure 7A*). Like the focal gelatin degradation patterns observed in AChR-poor regions of aneural clusters (*Figure 2A*), synaptic AChR clusters at the nerve-muscle contacts were found to be closely associated to, but not perfectly co-localized with, the sites of gelatin degradation, with only 17.15 ± 15% of area in AChR clusters to be associated with gelatin degradation in control co-cultures (*Figure 7C*). However, both gelatin degradation

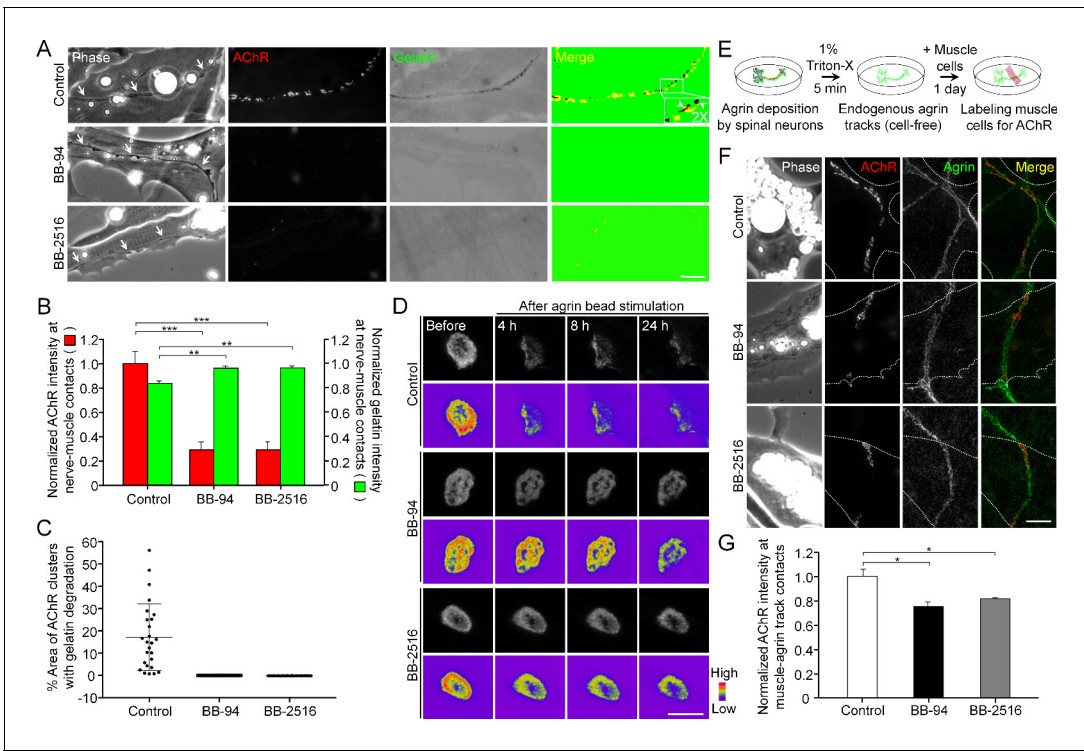

**Figure 7.** Pharmacological inhibition of MMP activity suppresses the assembly of synaptic AChR clusters and the disassembly of aneural AChR clusters. (A) Representative images showing the focal degradation of fluorescent gelatin and the formation of nerve-induced AChR clusters along the trails of nerve-muscle contacts (arrows) in control co-cultures, but not in co-cultures treated with MMP inhibitors. To better show the degree of colocalization, merge images were created by overlaying the binary images of AChR and fluorescent gelatin channels after thresholding. Arrowheads in the magnified inset indicate the spatial colocalization between synaptic AChR clusters and gelatin degradation (with only red signals). (B) Quantification on the effects of MMP inhibitors on fluorescent gelatin degradation and nerve-induced AChR clustering. n = 30 (Control), 32 (BB-94), and 38 (BB-2516) nerve-muscle pairs from 4 independent experiments. (C) Quantification on the degree of colocalization between synaptic AChR clusters and gelatin degradation in nerve-muscle contacts. n = 26 (Control), 36 (BB-94), and 36 (BB-2516) nerve-muscle pairs from 4 independent experiments. Data are represented as mean ± SD. (D) Representative time-lapse images showing the effects of MMP inhibitors on the dispersal of aneural AChR clusters upon agrin bead stimulation. 8-bit pseudo-color images highlight the change in fluorescence intensity of aneural AChR clusters over 24 hr in different conditions. (E) A schematic diagram illustrating the preparation of endogenous agrin tracks on ECL-coated substratum for inducing synaptic AChR clustering in cultured muscle cells. (F) Representative images showing the inhibitory effects of MMP inhibitors on synaptic AChR clustering induced by endogenous agrin tracks. The location of agrin tracks was visualized by agrin immunostaining and then confirmed if neurites were not found in the phase contrast images. Dotted lines outline the periphery of muscle cells. (G) Quantification on the effects of MMP inhibitors on synaptic AChR clustering induced by endogenous agrin tracks. n = 30 (Control), 27 (BB-94), and 22 (BB-2516) muscle cells from 3 independent experiments. Scale bars represent 10 μm. Data are represented as mean ± SEM, unless otherwise specified. One-way ANOVA with Dunnett's multiple comparisons test, *, **, *** represent p≤0.05, 0.01, and 0.001 respectively.

The online version of this article includes the following figure supplement(s) for figure 7:

**Figure supplement 1.** Agrin bead-induced AChR clustering is not affected by MMP inhibitors.

**Figure supplement 2.** Endogenous localization of MT1-MMP at aneural AChR clusters is elevated by agrin treatment.

and nerve-induced AChR clustering were effectively inhibited by BB-94 or BB-2516 (*Figure 7A and B*). Nerve innervation of skeletal muscles involves local signals to initiate the formation of synaptic AChR clusters and global signals to induce the dispersal of aneural AChR clusters (*Dai and Peng, 1998*). Thus, we further examined the involvement of MMP activity in the dispersal of aneural AChR clusters induced by latex beads coated with agrin, a heparan sulfate proteoglycan that induces post-synaptic differentiation. Local application of agrin-coated beads to cultured muscle cells is capable of inducing postsynaptic differentiation in a spatiotemporally controllable manner (*Lee et al., 2009*). In control cells, we detected a gradual dispersal of aneural AChR clusters upon agrin-bead stimulation for 24 hr; however, those clusters in agrin bead-contacted muscle cells were greatly stabilized by BB-94 or BB-2516 (*Figure 7D* and *Figure 7—figure supplement 1A*). To determine if agrin stimulation enhances the localization of MT1-MMP in aneural AChR clusters prior to their dispersal, we performed immunostaining experiments, in which we demonstrated an increase of endogenous MT1-MMP signals at aneural AChR clusters upon agrin stimulation (*Figure 7—figure supplement 2*). These data implicated the important role of MT1-MMP activity in regulating not only the assembly of synaptic AChR clusters at nerve-muscle contacts, but also the disassembly of aneural AChR clusters, at developing NMJs.

In contrast to nerve-induced AChR clustering, we noted that MMP inhibitors did not significantly affect the formation of AChR clusters induced by agrin beads (*Figure 7—figure supplement 1B and C*). As the latex beads were coated with a high amount of exogenous recombinant agrin, and they were placed on the top surface of cultured muscle cells, it remained unclear if MMP activity regulates synaptic AChR clustering induced by the physiological amount of agrin and in the presence of different ECM proteins. Therefore, we further developed an 'endogenous agrin track' assay (*Figure 7E*), by which we examined the effects of MMP inhibitors on synaptic AChR clustering in muscle cells cultured on ECL-coated substratum with endogenous agrin tracks. Using this assay, both BB-94 and BB-2516 were found to significantly inhibit the formation of synaptic AChR clusters induced by the endogenous agrin tracks (*Figure 7F and G*), consistent with our data on nerve-induced AChR clustering.

## Postsynaptic MT1-MMP is a regulator of NMJ development

To identify the postsynaptic functions of MT1-MMP in NMJ development, antisense MO was used to knock down the endogenous expression of MT1-MMP specifically in muscle cells and then co-cultured with wild-type (WT) spinal neurons. The efficiency of MO-mediated knockdown of endogenous MT1-MMP protein level in *Xenopus* embryos was validated by western blot analysis (*Figure 8—figure supplement 1*). Pre-existing and newly inserted AChR molecules were differentially labeled by α-bungarotoxin conjugated with different fluorophores according to our previously established protocol (*Lee et al., 2009*). In short, we first labeled all pre-existing AChR molecules with tetramethylrhodamine-conjugated α-bungarotoxin in muscle cells. After plating the spinal neurons for 1 day, the newly synthesized AChRs can then be labeled with Alexa Fluor 488-conjugated α-bungarotoxin. In control co-cultures, both pre-existing and newly inserted AChR signals were detected at the nerve-muscle contact sites (*Figure 8A*), suggesting that both receptor pools contribute to the assembly of nerve-induced synaptic AChR clusters. Importantly, MT1-MMP knockdown caused a significant reduction, but not a complete inhibition, of nerve-induced AChR clustering in the chimeric co-cultures of MT1-MMP MO muscles with WT neurons (*Figure 8A and B*). This could be explained by the partial knockdown of endogenous MT1-MMP protein level (*Figure 8—figure supplement 1*) and/or the recruitment of diffuse AChRs to the nerve-muscle contacts via MT1-MMP-independent processes. Intriguingly, both pre-existing and newly inserted AChRs at the nerve-muscle contacts were equally affected in the chimeric co-cultures of MT1-MMP MO muscles with WT neurons (*Figure 8C and D*). To rule out the possible off-target effects of MT1-MMP MO, we next performed rescue experiments by overexpressing MT1-MMP-mCherry (MT1-mCherry) in MT1-MMP knockdown (MT1-MO) muscle cells. In the chimeric co-cultures of MT1-MO + MT1-mCherry muscles with WT neurons, we found that the percentage of nerve-muscle contacts with AChR clusters was largely returned to the level comparable with the control groups, and both pre-existing and newly inserted AChRs were detected at the nerve-muscle contacts (*Figure 8*). These data suggested that postsynaptic MT1-MMP regulates both the surface delivery of newly synthesized AChR molecules and the redistribution/recruitment of pre-existing AChR molecules (from either aneural AChR clusters or diffuse AChRs

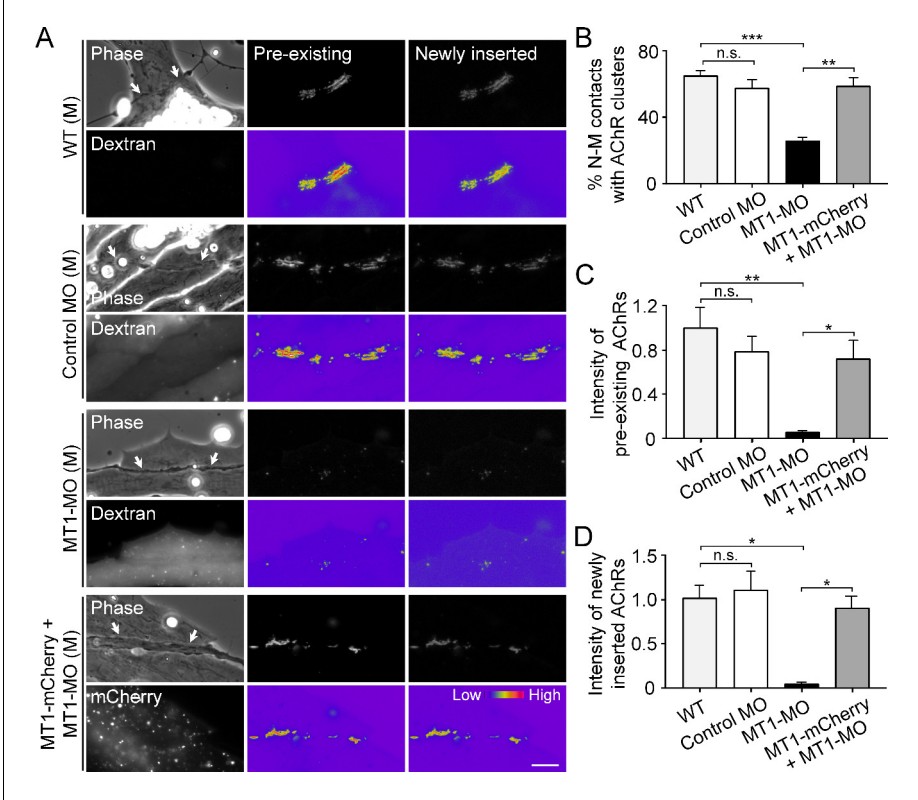

**Figure 8.** Postsynaptic MT1-MMP is required for nerve-induced AChR clustering. (**A**) Representative images showing the localization of pre-existing and newly inserted AChR clusters at nerve-muscle contacts (arrows) in control co-cultures (WT (M)) and in the chimeric co-cultures of wild-type neurons and muscle cells with control morpholino (Control MO (M)), MT1-MMP-MO (MT1-MO (M)), or MT1-MMP-mCherry and MT1-MMP-MO (MT1-mCherry + MT1-MO (M)) muscles. Fluorescent dextran indicates the muscle cells derived from MO-injected embryos. 8-bit pseudo-color images highlight the relative fluorescence intensity of pre-existing and newly inserted AChR signals. (**B**) Quantification on the percentage of nerve-muscle contacts with AChR clusters in control nerve-muscle co-cultures and in different chimeric co-cultures. n = 127 (WT), 120 (Control MO), 146 (MT1-MO), or 34 (MT1-mCherry + MT1-MO) nerve-muscle pairs from 3 independent experiments. (**C**) Quantification on the intensity of pre-existing AChR signals per unit length of nerve-muscle contacts in control co-cultures and in different chimeric co-cultures. n = 29 (WT), 30 (Control MO), 32 (MT1-MO), or 23 (MT1-mCherry + MT1-MO) nerve-muscle pairs from 4 independent experiments. (**D**) Quantification on the intensity of newly inserted AChR signals per unit length of nerve-muscle contacts in control co-cultures and in different chimeric co-cultures. n = 24 (WT), 23 (Control MO), 25 (MT1-MO), or 19 (MT1-mCherry + MT1-MO) nerve-muscle pairs from 3 independent experiments. Scale bar represents 10 μm. Data are represented as mean ± SEM. One-way ANOVA with Sidak's multiple comparisons test, *, **, *** represent p≤0.05, 0.01, and 0.001 respectively. n.s.: non-significant. The online version of this article includes the following figure supplement(s) for figure 8:

**Figure supplement 1.** Endogenous MT1-MMP protein level is effectively reduced by specific antisense morpholino oligonucleotides.

on muscle surface) during the assembly of nerve-induced postsynaptic specializations at developing NMJs.

## MT1-MMP regulates the recruitment of AChR molecules from aneural to synaptic clusters

To determine if aneural AChR clusters contribute to the formation of NMJs, we next performed laser-based photobleaching experiments to differentiate the recruitment of AChR molecules from aneural clusters versus diffuse AChRs for the assembly of nerve-induced synaptic AChR clusters (*Figure 9A*). Without photobleaching, AChR signals were prominently detected at the nerve-muscle contact sites. The formation of nerve-induced synaptic AChR clusters was accompanied by the

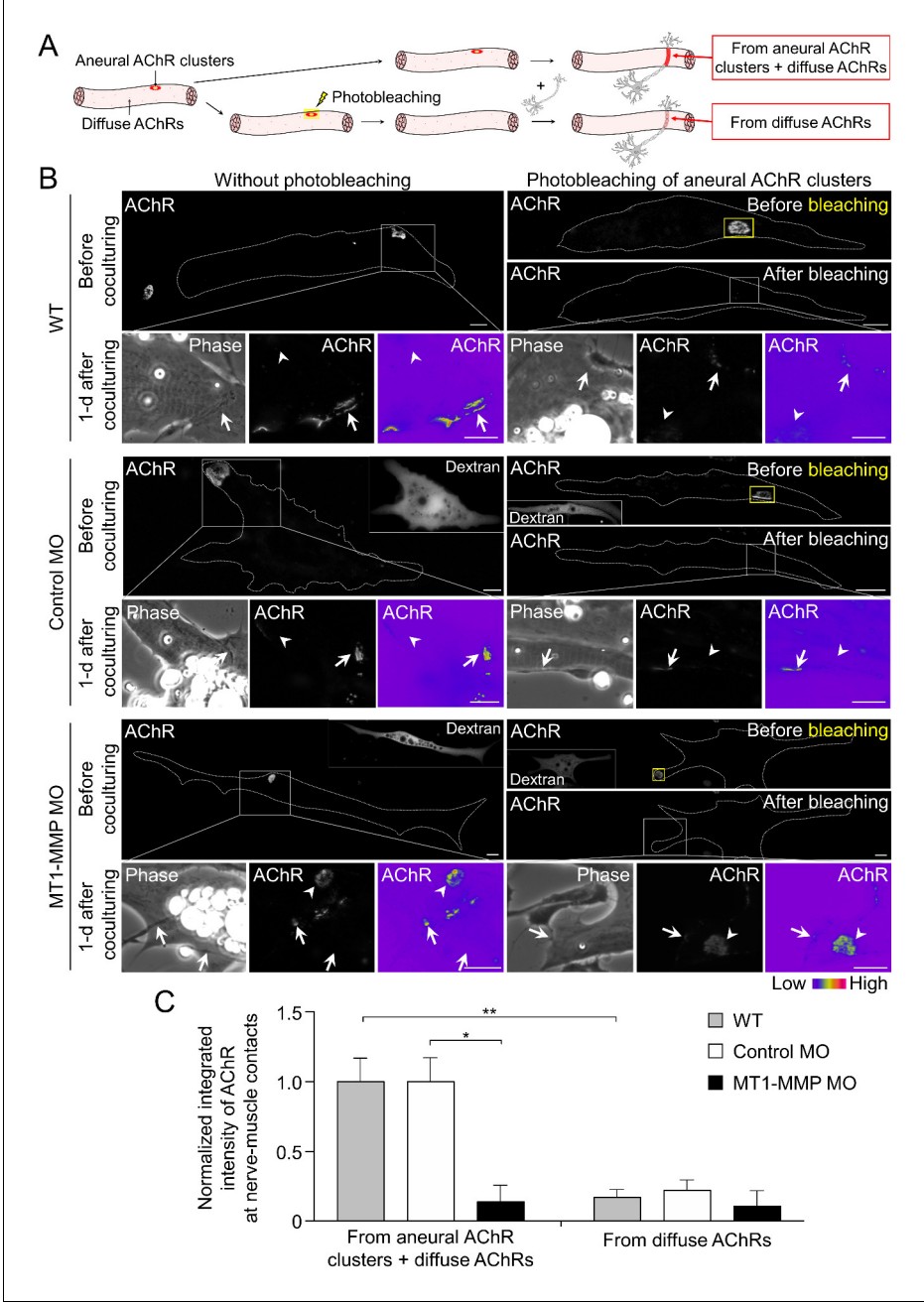

**Figure 9.** Postsynaptic MT1-MMP is required for the recruitment of aneural AChR clusters to the assembly of synaptic AChR clusters. (**A**) A schematic diagram illustrating the use of laser-based photobleaching approach to differentially identify the contribution of aneural AChR clusters and diffuse AChRs for the assembly of nerve-induced synaptic AChR clusters. (**B**) Representative images showing the differential contribution of aneural AChR clusters and diffuse AChRs to nerve-induced synaptic AChR clusters in control co-cultures (WT) and in the chimeric co-cultures of wild-type neurons and muscle cells with control MO or MT1-MMP MO. Yellow boxes (right panels) indicate the region of photobleaching. White boxes show a magnified view of nerve-muscle contacts in 1-d old co-cultures. Dotted lines highlight the periphery of muscle cells. Fluorescent dextran signals in the insets indicate the muscle cells with microinjected MO. 8-bit pseudo-color images highlight the relative fluorescence intensity of AChR signals. Arrows indicate synaptic AChR clusters at nerve-muscle contact sites. Arrowheads indicate the original location of aneural AChR clusters. (**C**) Quantification on the intensity of AChR signals at the nerve-muscle contacts in control co-cultures and in chimeric co-cultures either with or without photobleaching. The control groups without photobleaching indicate the contribution from aneural AChR clusters + diffuse AChRs, n = 15 (WT), 9 (Control MO), and 9 (MT1-MMP MO) nerve-muscle pairs from 3 independent experiments. The

*Figure 9 continued on next page*

*Figure 9 continued*

experimental groups with photobleaching of aneural AChR clusters indicate the contribution from diffuse AChRs only, n = 23 (WT), 13 (Control MO), and 8 (MT1-MMP MO) nerve-muscle pairs from 3 independent experiments. Scale bars represent 10 µm. Data are represented as mean ± SEM. Two-way ANOVA with Sidak's multiple comparison test, *, ** represent p≤0.05 and 0.01 respectively.

dispersal of aneural AChR clusters after 1 day in co-culture. In the experimental groups with photobleaching of aneural AChR clusters before co-culturing with the spinal neurons, we detected a significant reduction in AChR signals, which were contributed from diffuse AChRs only, at the nerve-muscle contacts in comparison to the control groups without photobleaching (*Figure 9B and C*). These results provided compelling evidence to support the hypothesis that AChR molecules from aneural clusters serve as a major source for the assembly of synaptic AChR clusters.

To further investigate the involvement of MT1-MMP in the recruitment of AChRs from aneural to synaptic clusters, we performed the same photobleaching experiments using chimeric co-cultures composed of either control MO or MT1-MMP MO muscle cells with WT spinal neurons (*Figure 9B*). In the chimeric co-cultures with control MO muscle cells, both aneural clustered and diffuse AChRs were recruited to the nerve-muscle contacts at a level comparable to the wild-type co-cultures. However, knockdown of MT1-MMP in muscles significantly suppressed the recruitment of AChR molecules from aneural to synaptic clusters, while the recruitment from diffuse AChRs was unaffected (*Figure 9C*). Consistent with the data above (*Figure 7D*), we found that knockdown of muscle MT1-MMP caused a higher stability of aneural AChR clusters against nerve-induced dispersal (arrowheads, lower left panels in *Figure 9B*). Interestingly, after photobleaching of aneural AChR clusters in the chimeric co-cultures with MT1-MMP MO muscles cells, the fluorescence intensity of photobleached aneural AChR clusters was partially recovered after 1 day in co-culture (arrowheads, lower right panels in *Figure 9B*). These results suggested that diffuse AChRs were constantly recruited to aneural AChR clusters that were stabilized by muscle MT1-MMP knockdown, presumably through the diffusion-trap mechanisms in which diffuse AChRs can be immobilized by molecular scaffolds associated with the stable aneural AChR clusters (*Geng et al., 2008*). Taken together, our data further supported that muscle MT1-MMP plays a dual-functional role in regulating the formation and dispersal of synaptic and aneural AChR clusters, respectively.

## Loss of MT1-MMP impairs AChR re-distribution and clustering at developing NMJs in vivo

To further investigate if MT1-MMP is required for NMJ development in vivo, we first performed immunostaining experiments to examine the localization of endogenous MT1-MMP at mature NMJs in rat soleus muscles. MT1-MMP was found to be highly enriched at the NMJs (*Figure 10A*). On the other hand, some immunostaining signals, which were partly contributed by intracellular MT1-MMP proteins, were also observed outside of the AChR cluster regions, further indicating that the precise control of MT1-MMP trafficking and surface insertion is required for mediating local ECM degradation at NMJs. The specificity of MT1-MMP antibody was validated by pre-incubating the antibody with recombinant MT1-MMP proteins, which caused a large reduction of immunostaining signals at the NMJs. To further determine whether MT1-MMP is postsynaptically localized, a denervation experiment on adult rat soleus muscle was performed (*Figure 10B*). In 4 days after sciatic nerve cut, we observed only a slight reduction in MT1-MMP signals at the NMJs of denervated muscles, suggesting that MT1-MMP is expressed primarily in postsynaptic muscles.

Next, we examined the patterns of AChR clusters in embryonic diaphragm muscles between WT and MT1-MMP-deficient mice, which was previously established (*Sakamoto and Seiki, 2009*; *Zhou et al., 2000*). Specifically, diaphragms from WT and MT1-MMP$^{-/-}$ mice at E13.5 and E18.5 were dissected and whole-mount stained for AChR and phrenic nerve. Aneural and synaptic AChR clusters were quantified based on whether the clusters were in direct contacts with the phrenic nerve branches. The endplate band width, which represents the zone of AChR cluster formation in the center of muscle fibers innervated by the phrenic nerve, was also measured. At E13.5, MT1-MMP$^{-/-}$ mice showed a significantly higher density of aneural AChR clusters than the age-matched WT mice, but the density of synaptic AChR clusters was largely unchanged at this stage (*Figure 10C-E*). At E18.5, we observed a significant reduction in the density of synaptic AChR clusters and endplate band

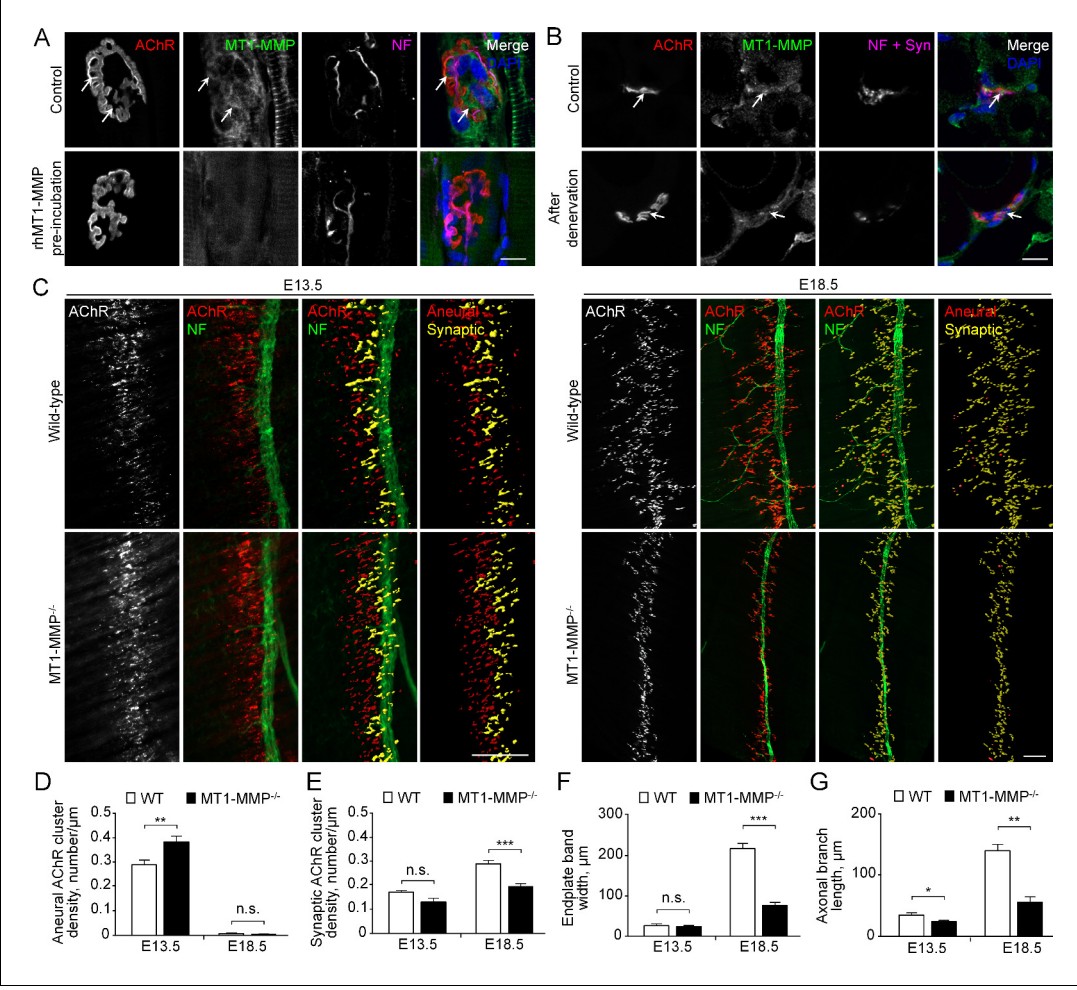

**Figure 10.** MT1-MMP is a regulator of NMJ development in vivo. (**A**) Representative confocal images showing the localization of endogenous MT1-MMP at perforations of synaptic AChR clusters (arrows, top panels) in longitudinal cryosections of adult rat soleus muscles. The specificity of MT1-MMP antibody was verified by pre-incubating anti-MT1-MMP primary antibody with recombinant human MT1-MMP protein (rhMT1-MMP, bottom panels). Neurofilament (NF) was used as a neuronal marker. (**B**) Representative confocal images showing the postsynaptic localization of MT1-MMP at NMJs as demonstrated by the surgical denervation experiment. Staining of AChR, MT1-MMP, and presynaptic marker (NF and synaptophysin (Syn)) was performed using cross sections from adult rat soleus muscles either without (top panels) or with sciatic nerve cut after 4 days (bottom panels). (**C**) Representative confocal images showing aneural versus synaptic AChR clusters in whole-mount diaphragms from wild-type (WT) control and MT1-MMP$^{-/-}$ mice at E13.5 (left panels) and E18.5 (right panels). Whole-mount tissues were stained for AChR and NF. The superimposed 3D reconstruction images were generated by z-stack images using Imaris. Synaptic AChR clusters (yellow) were identified when signals of AChR and NF overlapped with each other, whereas other AChR signals (red) represent aneural AChR clusters. (**D–G**) Quantification on the density of aneural (**D**) versus synaptic (**E**) AChR clusters, the width of end-plate bands (**F**), and the length of axonal branches (**G**) in diaphragm muscles between wild-type and MT1-MMP$^{-/-}$ mouse embryos at E13.5 and E18.5. n = 4 (E13.5, WT) and 5 (E13.5, MT1-MMP$^{-/-}$), 5 (E18.5, WT), and 3 (E18.5, MT1-MMP$^{-/-}$) embryos from three independent experiments. Scale bars represent 10 μm (**A**) or 100 μm (**C**). Data are represented as mean ± SEM. Two-way ANOVA with Sidak's multiple comparison test (**D and E**), Student's t-test (**F and G**), *, **, *** represent p≤0.05, 0.01, and 0.001 respectively. n.s.: non-significant.

width in MT1-MMP$^{-/-}$ mice compared to those in the WT mice (*Figure 10E-F*), indicating that synaptic AChR clusters formed in the knockout mice were immature with the altered number and density. Interestingly, the phrenic nerve in MT1-MMP$^{-/-}$ mice also exhibited abnormal phenotype with less axonal defasciculation and arborization in the diaphragm muscles (*Figure 10C and G*), suggesting

either the retrograde signaling of postsynaptic MT1-MMP or a possible role of neuronal MT1-MMP in regulating axonal growth and differentiation. Together, these findings further validated our in vitro data showing an important function of MT1-MMP in regulating the recruitment of aneural AChR clusters for the assembly of postsynaptic specializations at developing NMJs in vivo.

## Discussion

At developing NMJs, AChR pre-patterns can be formed in muscle fibers without motor innervation in vivo (*Lin et al., 2001*; *Yang et al., 2001*), suggesting that muscle-intrinsic mechanisms are sufficient to regulate the initial formation of aneural AChR clusters. In this study, we showed that the formation of topologically complex aneural AChR clusters in *Xenopus* muscle cells requires several key ECM proteins (*Figure 1*). As ECM proteins can be secreted and deposited by skeletal myofibers and other different cell types that constitute the synaptic basal lamina at developing NMJs, the initial formation of AChR clusters may not be solely mediated by muscle-intrinsic mechanisms.

The synaptic basal lamina is unique in composition, containing factors capable of inducing the assembly of synaptic specializations in both presynaptic and postsynaptic membranes, such as laminin-421, collagen IV/VI/XIII, heparan sulfate proteoglycans (e.g. agrin and perlecan), and other ubiquitously expressed ECM proteins (e.g. laminin-111, fibronectin and entactin/nidogen) (*Cescon et al., 2018*; *Chan et al., 2020*; *Latvanlehto et al., 2010*; *Singhal and Martin, 2011*). Among the different factors in the synaptic basal lamina, agrin plays the central role in the formation of postsynaptic specializations by activating the Lrp4/MuSK complex (*Kim et al., 2008*; *Zhang et al., 2008*). In this study, we observed extensive degradation of ubiquitously expressed ECM proteins, including gelatin and laminin-111, at AChR clusters (*Figure 2A-D*). A previous study suggested that MMP-3 is implicated for the removal of agrin from synaptic basal lamina at adult NMJs (*VanSaun and Werle, 2000*). In this study, however, the pattern and intensity of endogenous agrin tracks were largely unaffected by MMP inhibitors (*Figure 7F*), ruling out the possible proteolytic degradation of secreted agrin by MT1-MMP activity at developing NMJs. Therefore, we hypothesize that PLS-associated MMP activity focally degrades different ubiquitously expressed ECM proteins, which clear the zone at the nascent synaptic basal lamina for the deposition of other secreted synaptogenic factors (e.g. agrin or neuregulin) and/or dismantle the extracellular anchorage of AChR molecules on ECM proteins that allow aneural clusters to undergo dispersal and re-distribution upon nerve induction (*Figure 11*). Interestingly, agrin can serve as a mechanotransduction signal, which may transduce ECM and cellular rigidity signals to the Hippo pathway effector YAP, that requires both the Lrp4-MuSK signaling and integrin-focal adhesion signaling in liver cancer cells (*Chakraborty et al., 2017*). Importantly, muscle YAP mutation was found to impair the formation of NMJs and to inhibit NMJ regeneration after nerve injury (*Zhao et al., 2017*). Therefore, the possible crosstalk between ECM-integrin and agrin-Lrp4-MuSK signaling pathways in regulating the postsynaptic differentiation at NMJs remains to be investigated.

PLSs are highly dynamic actin-rich organelles found in transformed fibroblasts and many other motile or invasive cells. We detected several typical core and cortex markers of PLSs that were highly concentrated at perforated AChR clusters (*Figure 1—figure supplement 2*), further confirming the presence of PLSs in primary myotomal muscle cultures. Intriguingly, PLS cortex proteins, like vinculin and talin, were enriched not only at the perforated regions, but also at the periphery, of aneural AChR clusters (*Figure 1—figure supplement 2B*). At those sites, some MT1-MMP-mCherry signals were captured (*Figure 5D*), and some MT1-MMP-pHluorin signals were spatially recovered after photobleaching (*Figure 6F*). These data suggested that PLS cortex proteins may capture MT1-MMP-containing vesicles and facilitate their surface insertion at the perforation and periphery of AChR clusters. It is important to note that both signals of MT1-MMP immunostaining (*Figure 3A*) and MT1-MMP-mCherry overexpression (*Figure 5*) were not selectively localized to aneural AChR clusters, as both surface and intracellular MT1-MMP at the vesicular compartments were highlighted. In contrast, the localization patterns of MT1-MMP-pHluroin at a low expression level (*Figure 6A*) and fluorescent gelatin degradation (*Figure 2A*) were primarily restricted to the perforated AChR clusters. These results suggested that the vesicular trafficking and surface insertion of MT1-MMP is precisely controlled to mediate ECM degradation during NMJ formation.

Microtubules are important cytoskeletal structures for intracellular transport of proteins. EB1, a key regulator of the plus-end-tracking protein complex, functions to modulate microtubule dynamics

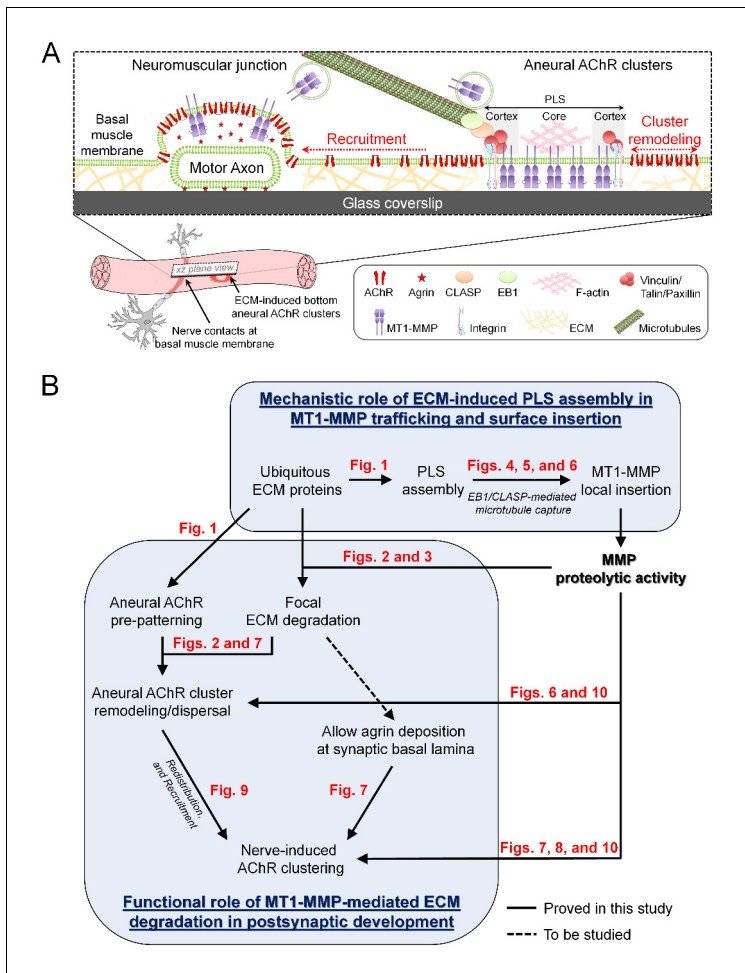

**Figure 11.** Working model on site-directed vesicular trafficking and surface delivery of MT1-MMP to regulate AChR cluster formation and remodeling. (**A**) Schematic diagram showing the proposed mechanisms underlying site-directed MT1-MMP trafficking and surface insertion for the regulation of focal ECM degradation and the recruitment of ECM-induced bottom aneural AChR clusters to nerve-induced synaptic clusters at developing NMJs. (**B**) Logical flow diagram proposing key events in the mechanistic role of ECM-induced PLS assembly in MT1-MMP trafficking and surface insertion, and the functional role of MT1-MMP-mediated ECM degradation in postsynaptic development. The relevant data presented in the main figures that support each of the individual proposed events are highlighted in red.

and interaction with other intracellular organelles. Although endogenous EB1 comets can be found in the entire cell, they are highly enriched at the sites specialized for efficient coordination of cargo transport and release, such as focal adhesions and NMJs (*Noordstra and Akhmanova, 2017*; *Schmidt et al., 2012*). Direct interactions between EB1 and other microtubule plus-end-tracking proteins (CLASPs and CLIP-170) are implicated in the 'recruitment mechanism', by which EB1 functions to search-and-capture post-Golgi vesicles and to spatially capture microtubules at the cell cortex (*Vaughan, 2005*). LL5β, one of the PLS cortex proteins, is localized at the postsynaptic membrane to serve as a corral for AChRs within the clusters (*Kishi et al., 2005*). Interestingly, LL5β and CLASP1 interact with microtubules for directing the vesicular trafficking of different proteins to the postsynaptic membrane of NMJs (*Basu et al., 2015*) and to the focal adhesions (*Stehbens et al., 2014*). In this study, antisense MO-mediated knockdown of endogenous *Xenopus* CLASP, the homolog of mammalian CLASP1, in cultured muscle cells significantly reduced the local capture of microtubules at AChR clusters, as reflected by the increased average speed of EB1-GFP comets within AChR regions (*Figure 4C*). It was also noted that microtubule plus-end-tracking proteins are capable to hop on and off from microtubule ends (*Dragestein et al., 2008*), thus we cannot exclude the

possibility of EB1-GFP signal recovery at the photobleached region that may be partly contributed by the exchange of EB1-GFP molecules between the free and microtubule-bound pools. Given that MT1-MMP surface insertion at aneural AChR clusters was significantly inhibited in CLASP-MO muscle cells (*Figure 6H*), we therefore hypothesize that EB1-mediated microtubule capturing mechanisms can be regulated by PLSs at both cortex (adhesion molecule-based) and core (actin-based) domains via CLASP-dependent and -independent manner respectively, which is in agreement with other previous studies demonstrating the involvement of microtubule-dependent vesicular transport in AChR clustering (*Basu et al., 2015*; *Marchand et al., 2002*; *Schmidt et al., 2012*).

It has long been suggested that innervation of the skeletal muscle involves local signaling to initiate the formation of synaptic AChR clusters at the nerve-muscle contacts, and global signaling to induce the dispersal of aneural AChR clusters (*Dai and Peng, 1998*). However, a compelling evidence supporting that aneural AChR clusters serve as source for synaptic AChR clusters at developing NMJs is still lacking. In this study, we used laser-based photobleaching approach to demonstrate the differential contribution of AChR molecules from aneural clusters and diffuse AChRs for the assembly of nerve-induced synaptic AChR clusters (*Figure 9*). In agreement with previous studies (*Anderson and Cohen, 1977a*; *Cohen et al., 1987*), most nerve contacts in *Xenopus* primary cultures are located at either the basal surface or the lateral side of muscle cells, which are influenced by MMP-mediated proteolytic degradation of ECM proteins. Importantly, we demonstrated that the recruitment of AChRs from aneural clusters, but not diffuse AChRs, is mediated by MT1-MMP-dependent processes. While MT1-MMP knockdown caused a significant inhibition in synaptic AChR clustering at the nerve-muscle contacts, it is intriguing to note that diffuse AChRs were constantly recruited to aneural AChR clusters that were stabilized by MT1-MMP knockdown (*Figure 9B*). This recruitment may be mediated by the diffusion-trap mechanism, in which diffuse AChRs can be tethered to molecular scaffolds associated with the stable aneural AChR clusters (*Geng et al., 2008*). Therefore, we hypothesize that MT1-MMP facilitates the disassembly of extracellular (e.g. ECM proteins) and/or intracellular (e.g. cytoskeletal proteins) molecular scaffolds that are tightly associated with aneural AChR clusters, such that AChR molecules from dispersing aneural clusters can be recruited to synaptic clusters in response to nerve induction. In a minority of cases, some aneural and nerve-induced AChR clusters can be found at the top muscle surface, where no ECM proteins are directly involved. Particularly, those top aneural AChR clusters are spontaneously formed with possibly no physiological relevance, whether MT1-MMP regulates the formation and remodeling of ECM-independent top AChR clusters remains unknown. Collectively, our present study further supported that postsynaptic MT1-MMP plays a dual-functional role in regulating the formation and dispersal of synaptic and aneural AChR clusters, respectively, via focal ECM degradation.

MT1-MMP is the best characterized and the most prevalent isoform of the large MMP family, including membrane-type or soluble MMPs. MT1-MMP deficiency causes myogenic impediments and central nucleation of myofibers, which are typically found in muscular dystrophy (*Ohtake et al., 2006*). Besides, MT1-MMP knockout mice develop multiple abnormalities and die between 50–90 postnatal days (*Holmbeck et al., 1999*), suggesting that the possibility of functional redundancy by other MMP isoforms is low. In this study, we used mutant mice lacking MT1-MMP in all cells and observed that the density of aneural AChR clusters was significantly higher than that in wild-type mice at E13.5 (*Figure 10D*), consistent with the in vitro findings on the involvement of MT1-MMP in the recruitment of AChR molecules from aneural to synaptic AChR clusters (*Figure 9*). However, at E18.5, the density of aneural AChR clusters between MT1-MMP$^{-/-}$ and wild-type mice was comparable. While aneural AChR clusters in MT1-MMP$^{-/-}$ mice were not recruited to synaptic AChR clusters, they were subjected to the regular metabolic turnover/degradation, leading to the disappearance of aneural AChR clusters at E18.5. This is supported by previous studies showing that the half-life of extra-synaptic AChRs is only about 19 hr, compared with 8–14 days for synaptic AChRs at mature NMJs (*Chang and Huang, 1975*; *Steinbach et al., 1979*). On the other hand, reduced synaptic AChR density in MT1-MMP$^{-/-}$ mice (*Figure 10E*) is attributed by (1) significantly declined recruitment of AChR molecules from aneural clusters, and (2) reduced amount of newly inserted AChRs into the postsynaptic regions. These data provided a solid validation of our in vitro findings that demonstrate the important role of MT1-MMP in regulating the recruitment of pre-existing and newly inserted AChRs to the nerve-muscle contact sites (*Figure 8*). It is also important to note that the phrenic nerve in MT1-MMP$^{-/-}$ mice showed a significant reduction of axonal growth and arborization in the

diaphragm muscles at both E13.5 and E18.5 (*Figure 10G*). This phenotype on defects in axonal growth could be explained by the retrograde signaling initiated by postsynaptic MT1-MMP, which could mediate the extracellular cleavage of Lrp4 to generate ecto-Lrp4 fragments that activate agrin signaling in trans (*Wu et al., 2012*). In fact, Lrp4 is also considered as a retrograde signal to induce presynaptic differentiation in vivo (*Yumoto et al., 2012*). Alternatively, postsynaptic MT1-MMP could also mediate the proteolytic conversion of proBDNF to mature BDNF (*Je et al., 2012*) that promotes neuronal survival and outgrowth in cultured neurons (*Peng et al., 2003*). Nevertheless, by using the whole-body MT1-MMP knockout mice in the present study, we cannot rule out the possible secondary effects on postsynaptic AChR clustering and remodeling that are contributed by the axonal growth and presynaptic differentiation defects in the absence of neuronal MT1-MMP. Therefore, our future studies plan to develop muscle-specific inducible MT1-MMP knockout animal models to further understand how muscle MT1-MMP controls the formation and maturation of NMJs in vivo. Particularly, it would be of great interest to address whether synaptic maturation, as reflected by the plaque-to-pretzel topological transition of AChR clusters during the early postnatal stages in mice, is critically driven by neuronal or muscle MT1-MMP.

From a clinical perspective, increased serum levels of several MMPs have been identified in patients with myasthenia gravis, an autoimmune neuromuscular disease (*Helgeland et al., 2011*). As inhibiting MMP activity could greatly stabilize aneural AChR clusters against spontaneous and nerve-induced dispersal (*Figure 2I* and *Figure 7D*), it would be interesting to test if manipulating muscle MT1-MMP expression level or its activity can suppress the pathogenic action in causing NMJ disassembly using our recently established *Xenopus* cell-based assay for investigating the pathogenesis of myasthenia gravis (*Chan et al., 2017*; *Yeo et al., 2015*). Given the physiological and pathophysiological roles of MMPs at the NMJs, understanding the cell biology of MT1-MMP trafficking and surface delivery in this study may provide additional insight into the regulation of synaptic function and dysfunction by modulating MMP trafficking and activity in neuromuscular development and diseases.

# Materials and methods

## Key resources table

| Reagent type (species) or resource | Designation | Source or reference | Identifiers | Additional information |
|---|---|---|---|---|
| Gene (*X. laevis*) | MT1-MMP | Xenbase | XB-GENE-485647 | |
| Genetic reagent (*M. musculus*) | MT1-MMP⁻/⁻ | Professor Zhongjun Zhou (HKU) *Sakamoto and Seiki, 2009* | | - |
| Cell line (*M. musculus*) | C2C12, Mouse immortalized myoblast | ATCC | Cat#: CRL-1772, RRID:CVCL-0188 | From ATCC; Cell identity has been confirmed by STR profiling and cell line was found to be free of *Mycoplasma*. |
| Biological sample (*X. laevis*) | Primary muscle and neuronal culture | *Xenopus* 1 | | |
| Biological sample (Rat) | Rat soleus muscle | Laboratory Animal Unit (HKU) | - | Sprague Dawley |
| Antibody | Anti-β-tubulin | Developmental Studies Hybridoma Bank | Cat#: 6G7-s, RRID:AB_528497 | IF (1:1000) |
| Antibody | Anti-α-tubulin | Sigma-Aldrich | Cat#: T6074, RRID:AB_477582 | WB (1:2000) |
| Antibody | Anti-Agrin | Developmental Studies Hybridoma Bank | Cat#: 6D2, RRID:AB_528071 | IF (1:100) |

*Continued on next page*

*Continued*

| Reagent type (species) or resource | Designation | Source or reference | Identifiers | Additional information |
|---|---|---|---|---|
| Antibody | Anti-ADF/cofilin | Dr James Bamburg (Colorado State University) *Lee et al., 2009* | | IF (1:500) |
| Antibody | Anti-β-actin | Sigma-Aldrich | Cat#: A2228, RRID:AB_476697 | WB (1:1000) |
| Antibody | Anti-Cortactin | Santa Cruz Biotechnology | Cat#: sc11408, RRID:AB_2088281 | IF (1:500) |
| Antibody | Anti-CLASP1 | Abcam | Cat#: ab85919 | WB (1:250) |
| Antibody | Anti-EB1 | BD Biosciences | Cat#: 610534, RRID:AB_397891 | IF (1:100) |
| Antibody | Anti-Laminin | Thermo Fisher Scientific | Cat#: PA516287, RRID:AB_10985513 | IF (1:100) |
| Antibody | Anti-MT1-MMP | Merck Millipore | Cat#: MAB3328, RRID:AB_570599 | IF (1:100 or 1:50) |
| Antibody | Anti-MT1-MMP | Merck Millipore | Cat#: AB6005, RRID:AB_10618742 | WB (1:1000), a gift from Dr Timothy Gomez (University of Wisconsin-Madison) |
| Antibody | Anti-MT1-MMP | Santa Cruz Biotechnology | Cat#: sc-373908 | IF (1:100) |
| Antibody | Anti-NF200 | Sigma-Aldrich | Cat#: N4142, RRID:AB_477272 | IF (1:500 or 1:1000) |
| Antibody | Anti-Paxillin | Biolegend | Cat#: 624001, RRID:AB_2300550 | IF (1:100) |
| Antibody | Anti-p34-Arc | EMD Millipore | Cat#: 07–227, RRID:AB_310447 | IF (1:100) |
| Antibody | Anti-Synaptobrevin 1 | Synaptic Systems | Cat#: 104002, RRID:AB_887807 | IF (1:300) |
| Antibody | Anti-Synaptophysin | Abcam | Cat#: ab32127, RRID:AB_2286949 | IF (1:1000) |
| Antibody | Anti-Talin | Biolegend | Cat#: T3287, RRID:AB_477572 | IF (1:100) |
| Antibody | Anti-Vinculin | Sigma-Aldrich | Cat#: V4505, RRID:AB_477617 | IF (1:100) |
| Recombinant DNA reagent | GFP-EB1 | Addgene | RRID:Addgene_39299 | |
| Recombinant DNA reagent | MT1-MMP-mCherry | Dr Cheng-Han Yu (HKU) | | |
| Recombinant DNA reagent | MT1-MMP-pHluorin | Dr Cheng-Han Yu (HKU) | | |
| Sequence-based reagent | Control morpholino | Gene Tools | Standard control oligo | CCTCTTACCTCAGTTACAATTTATA |
| Sequence-based reagent | MT1-MMP morpholino | Gene Tools | Custom morpholino | CCAGGCTGCTCTCAGAGGCTCCATC |
| Sequence-based reagent | CLASP morpholino | Gene Tools | Custom morpholino | GCCAGTAGTCCATTCCCTGTTCCAT |
| Peptide, recombinant protein | α-Bungarotoxin | Thermo Fisher Scientific | Cat#: B1601 | |

*Continued on next page*

Continued

| Reagent type (species) or resource | Designation | Source or reference | Identifiers | Additional information |
|---|---|---|---|---|
| Peptide, recombinant protein | Alexa Fluor 488 Phalloidin | Thermo Fisher Scientific | Cat#: A12379 | |
| Peptide, recombinant protein | Alexa Fluor 647 conjugate-α-bungarotoxin | Thermo Fisher Scientific | Cat#: B35450 | |
| Peptide, recombinant protein | Alexa Fluor 488 conjugate-α-bungarotoxin | Thermo Fisher Scientific | Cat#: B13422 | |
| Peptide, recombinant protein | Corning Collagen I | Fisher Scientific | Cat#: C354249 | |
| Peptide, recombinant protein | Dextran, Alexa Fluor 488 | Thermo Fisher Scientific | Cat#: D22910 | |
| Peptide, recombinant protein | Dextran, Alexa Fluor 546 | Thermo Fisher Scientific | Cat#: D22911 | |
| Peptide, recombinant protein | ECL cell attachment matrix | Merck Millipore | Cat#: 08–100 | |
| Peptide, recombinant protein | Gelatin | Sigma-Aldrich | Cat#: G1393 | |
| Peptide, recombinant protein | Gelatin From Pig Skin, Fluorescein Conjugate | Thermo Fisher Scientific | Cat#: G13187 | |
| Peptide, recombinant protein | Gelatin From Pig Skin, Oregon Green 488 Conjugate | Thermo Fisher Scientific | Cat#: G13186 | |
| Peptide, recombinant protein | Laminin from Engelbreth-Holm-Swarm murine sarcoma | Sigma-Aldrich | Cat#: L2020 | |
| Peptide, recombinant protein | Poly-D-lysine hydrobromide | Sigma-Aldrich | Cat#: P1024 | |
| Peptide, recombinant protein | Recombinant human MT1-MMP | R and D Systems | Cat#: 918-MP-010 | |
| Peptide, recombinant protein | Tetramethylrhodamine α-bungarotoxin | Thermo Fisher Scientific | Cat#: T1175 | |
| Commercial assay or kit | Alexa Fluor 488 Tyramide SuperBoost Kit, goat anti-mouse IgG | Thermo Fisher Scientific | Cat#: B40941 | |
| Commercial assay or kit | Alexa Fluor 594 Tyramide SuperBoost Kit, goat anti-rabbit IgG | Thermo Fisher Scientific | Cat#: B40944 | |
| Chemical compound, drug | BB-94 | ApexBio | Cat#: A2577-S | |

*Continued on next page*

*Continued*

| Reagent type (species) or resource | Designation | Source or reference | Identifiers | Additional information |
|---|---|---|---|---|
| Chemical compound, drug | Collagenase from Clostridium histolyticum | Sigma-Aldrich | Cat#: C98191G | |
| Chemical compound, drug | Leibovitz's L-15 medium | Sigma-Aldrich | Cat#: L4386 | |
| Chemical compound, drug | Marimastat/ BB-2516 | ApexBio | Cat#: A4049 | |
| Chemical compound, drug | Sodium borohydride | Sigma-Aldrich | Cat#: 452882 | |
| Software, algorithm | ZEN 2.3 Blue Edition | Carl Zeiss Microscopy | | |
| Software, algorithm | Prism 7 | GraphPad Software | | |
| Software, algorithm | ImageJ 1.52a (Java 1.8.0_66 (64-bit)) | NIH | https://imagej.nih.gov/ij/ | |
| Software, algorithm | Imaris 7 | Oxford Instruments | | |
| Software, algorithm | MetaMorph 7.8.2 | Molecular Devices | | |
| Software, algorithm | μmanager 1.4 | Open Imaging | | |
| Other | CryoStar NX50 Cryostat | Thermo Fisher Scientific | | |
| Other | DAPI-Fluoromount-G | Electron Microscopy Sciences | Cat#: 17984–24 | |
| Other | Fluoromount-G | Thermo Fisher Scientific | Cat#: 00-4958-02 | |
| Other | Nanoject II auto-nanoliter injector | Drummond Scientific | Cat#: 3-000-206A | |

### *Xenopus* embryonic muscle and nerve-muscle cultures

Adult *Xenopus laevis* animals were purchased from *Xenopus* 1. *Xenopus* oocytes were fertilized in vitro, and the embryos were raised in Holtfreter's solution (vol/vol; 60 mM NaCl, 0.6 mM KCl, 0.9 mM $CaCl_2$, 0.2 mM $NaHCO_3$, pH 7.4) at 22°C. 20–100 pg of DNA constructs encoding MT1-MMP-pHluorin, MT1-MMP-mCherry (a gift from Dr Cheng-Han Yu, The University of Hong Kong) or EB1-GFP (Addgene, 39299) were microinjected into 1 cell stage *Xenopus* embryos using an oocyte injector, Nanoject (Drummond Scientific). GFP- or mCherry-expressing embryos were screened for primary culture preparation, as previously described (*Lee et al., 2009*; *Peng et al., 1991*). In short, dorsal parts of wild-type or microinjected embryos at Nieuwkoop and Faber stage 19–22 were dissected. After the enzymatic digestion by collagenase (Sigma, C98191G), myotomal tissues and neural tubes were isolated, followed by dissociation with calcium-magnesium-free solution. The dissociated cells were plated on either coverslips or glass-bottom dishes coated with a mixture of cell attachment substrate, entactin-collagen IV-laminin (ECL) (Millipore, 08–100), or specific ECM proteins including laminin (Sigma, L2020), collagen I (Fisher Scientific, C354249), and gelatin (Sigma, G1393) at 10 μg/ml, or poly-D-lysine (Sigma, P1024) at 100 μg/ml. Cells were maintained in culture medium containing 87% Steinberg's solution (vol/vol; 60 mM NaCl (Sigma, S3014), 0.67 mM KCl (Sigma, P5405), 0.35 mM $Ca(NO_3)_2$ (Sigma, 202976), 0.83 mM $MgSO_4$ (Sigma, M2773), 10 mM HEPES (Sigma, H3375), pH 7.4), 10% Leibovitz's L-15 medium (Sigma, L4386) (vol/vol), 1% fetal bovine serum (Gibco, 10270) (vol/vol), 1% penicillin/streptomycin (Thermo Fisher Scientific,

15140122) (vol/vol) and 1% gentamicin sulfate (Thermo Fisher Scientific, 15750060) (vol/vol). Muscle cells were kept at 22°C for at least 24 hr to allow cell attachment and development of aneural AChR cluster before treatments, if any. To make nerve-muscle co-cultures, dissociated spinal neurons were plated into 2 d-old muscle cultures and grew for 1 day before imaging. Most of the nerve-contacted sites can be found in the basal membrane of muscle cells on ECM-coated substrates, where the growth cones of spinal neurons are able to crawl and migrate under the muscle cells (*Anderson and Cohen, 1977a*; *Anderson et al., 1977b*). All experiments involving *Xenopus* frogs and embryos were carried out in accordance with the approved protocol (#4627–18) by the Committee on the Use of Live Animals in Teaching and Research (CULATR) of The University of Hong Kong.

## C2C12 myotube culture

Authenticated C2C12 cell line was purchased from ATCC. Cell identity has been confirmed by STR profiling and cell line was found to be free of Mycoplasma. Cells were cultured in Dulbecco's Modified Eagle's Medium (DMEM) containing 20% fetal calf serum supplemented with penicillin/streptomycin. Only cells with low passage (<5) were used in this study. After trypsinization, cells were cultured on glass coverslips coated with 1 mg/ml fluorescent gelatin. To induce myoblast fusion, the culture medium was replaced with the fusion medium containing 2% horse serum (Thermo Fisher Scientific, 16050130) in DMEM supplemented with penicillin/streptomycin.

## MT1-MMP knockout mice

MT1-MMP$^{+/-}$ mice in C57BL/6 background were obtained from Motoharu Seiki (Kanazawa University, Japan). In short, exons 1–5 encoding the catalytic domain of MT1-MMP were substituted by LacZ coding sequence (*Sakamoto and Seiki, 2009*). MT1-MMP$^{+/-}$ mice were backcrossed against C57BL/6 background mice at least 12 times. MT1-MMP$^{+/-}$ mice were mated to produce homozygous MT1-MMP$^{-/-}$ mice, which were confirmed by genotyping. All animal experiments were carried out in accordance with the approved protocol (#5149–19) by the Committee on the Use of Live Animals in Teaching and Research (CULATR) of The University of Hong Kong.

## Rat surgery, tissue harvest, and cryopreservation

6-week-old rats received anesthesia by intraperitoneal injection of 130 mg/kg ketamine with 13 mg/kg xylazine. After anesthesia, the left sciatic nerve was exposed with a lateral longitudinal straight incision from the greater trochanter to mid-thigh, and then by blunt dissection between the quadriceps femoris and biceps femoris. After clearing the surrounding connective tissues, the left sciatic nerve was cut with scissors. A sham operation was performed, where the sciatic nerve was exposed, and then the skin was closed immediately afterwards. After the surgery, the rats were returned to the cages in a pathogen-free environment, in a 12 hr light/dark cycle and with water and food ad libitum. Rats were euthanized 4 days after surgery with an overdose of pentobarbital (300 mg/kg). Then, soleus muscles were harvested from the control and denervated rats. After the fixation of the harvested muscles with 2% paraformaldehyde for 30 min, the tissues were then covered with tissue-tek O.C.T. compound (Sakura), and frozen with methanol in liquid nitrogen. Serial longitudinal and cross sections (20 μm thick) were cut in a cryostat microtome at −15°C (CryoStar NX50 Cryostat, Thermo Scientific). All animal experiments were carried out in accordance with the approved protocol by the Committee on the Use of Live Animals in Teaching and Research (CULATR) of The University of Hong Kong.

## Morpholino-mediated knockdown of endogenous proteins

Knockdown of endogenous proteins in *Xenopus* was achieved by embryonic injection of antisense morpholino oligonucleotides (MO, Gene Tools), which bind to the target mRNA sequence to block its protein translation. The following MO sequences were used in this study: *Xenopus* MT1-MMP MO: 5'-CCAGG CTGCT CTCAG AGGCT CCATC-3', *Xenopus* CLASP MO: 5'-GCCAG TAGTC CA TTC CCTGT TCCAT-3', and standard control MO: 5'-CCTCT TACCT CAGTT ACAAT TTATA-3'. To visualize the presence of MO in the microinjected embryos, MOs were co-injected with Alexa Fluor 488- or Alexa Fluor 546-conjugated dextran (Thermo Fisher Scientific, D22910 or D22911) as a cell lineage tracer. The effectiveness of MO-mediated knockdown of endogenous proteins was validated by Western blot analyses.

## Pharmacological treatment

For the experiments studying the effect of MMP inhibitors on the formation of aneural AChR clusters, 5 μM BB-94 (ApexBio, A2577) or 10 μM Marimastat/BB-2516 (ApexBio, A4049) was added in the culture medium before muscle cell plating. For the experiments studying the nerve-induced or agrin bead-induced AChR clusters, BB-94 or BB-2516 was added to the muscle cultures about 1 hr before adding the spinal neurons or agrin beads.

## Live-cell staining

Surface AChRs were labeled with 0.1 μM tetramethylrhodamine-, Alexa Fluor 488-, or Alexa Fluor 647-conjugated α-bungarotoxin (Thermo Fisher Scientific, T1175, B13422, or B35450) for 45 min in culture medium, followed by extensive washing with culture medium. To differentially label the pre-existing and newly-inserted pools of AChRs (*Lee et al., 2009*), the pre-existing AChRs were first labeled with tetramethylrhodamine-conjugated α-bungarotoxin for 45 min, followed by saturating all unlabeled surface AChRs with a high dose (6 μM) of unconjugated α-bungarotoxin (Thermo Fisher Scientific, B1601) for 30 min and then washed extensively with culture medium for at least 3 times. After 1 day, newly inserted AChRs were labeled with 1 μM Alexa Fluor 488-conjugated α-bungarotoxin. Glass coverslips with live cultured cells were then mounted on custom-made sealed chambers containing culture medium for live-cell imaging.

## Fluorescent gelatin degradation assay

Glass coverslips (Fisher Scientific, 12-545-82) or glass-bottom dishes (MatTek, P35G-1.5–14 .C-GRID) were coated with 1 mg/ml Oregon Green 488-gelatin or FITC-gelatin (Thermo Fisher Scientific, G13186 or G13187) for 10 min and followed by cross-linking with 0.5% glutaraldehyde (Sigma, G5882) in PBS for 15 min. After that, gelatin-coated coverslips or glass-bottom dishes were treated with 5 mg/ml $NaBH_4$ (Sigma, 452882) for 3 min. Dissociated cells were plated on fluorescent gelatin-coated coverslips or glass-bottom dishes and were maintained for different time periods with or without treatment, as specified. All images were taken using the same acquisition settings across different experimental groups. The extent of gelatin degradation was quantified using ImageJ (NIH), by which the mean intensity of fluorescent gelatin signals within the regions of aneural AChR clusters or nerve-muscle contact sites was measured, and then normalized with the mean intensity at the adjacent region in the same cell (except in MT1-MMP-mCherry overexpression experiments, the mean intensity at the adjacent region without cell attachment was used for normalization).

## Endogenous agrin track assay

Dissociated *Xenopus* spinal neurons were first plated on ECL-coated coverslips and cultured for 1 day. Spinal neurons were then digested and removed with 1% Triton X-100 in PBS for 5 min, followed by extensive washing with PBS for at least 5 times for 5 hr. For the experimental groups, pharmacological agents were added into the culture medium before plating the dissociated muscle cells. After 1 day in culture, the location of agrin tracks was visualized by agrin immunostaining (1:100; DSHB, 6D2) and then confirmed if neurites were not found in the phase contrast images.

## Fixation and immunostaining of cells and tissues

Cultured cells were either fixed with 4% paraformaldehyde (Thermo Fisher Scientific, 28908) in PBS for 15 min, followed by permeabilization with 0.5% Triton X-100 in PBS for 10 min or fixed with −20°C 95% ethanol for 5 min. For EB1 immunostaining, muscle cells were fixed with 90% methanol containing 50 mM EGTA at −20°C for 15 min, followed by further fixation with 4% PFA at room temperature for 10 min. The fixed cells were washed extensively with PBS for at least 3 times, followed by blocking overnight with 2% Bovine Serum Albumin (Sigma, A9418) at 4°C. Cells were incubated with primary antibodies, including β-tubulin (1:1000; DSHB, 6G7-s),cortactin (1:500; Santa Cruz Biotechnology, sc11408), paxillin (1:100; Biolegend, 624001), p34-Arc (1:100; Millipore, 07–227), talin (1;100; Sigma, T3287), vinculin (1:100; Sigma, V4504), laminin (1:100; Thermo Fisher Scientific, PA516287), EB1 (1:100; BD Biosciences, 610534), Synaptobrevin/VAMP1 (1:300; Synaptic Systems,104002), MT1-MMP (1:50; Millipore, MAB3328), or ADF/cofilin (1:500; a gift from Dr James Bamburg, Colorado State University) at room temperature for 2 hr, followed by fluorophore-conjugated secondary antibodies (Thermo Fisher Scientific) for 45 min. For MT1-MMP immunostaining,

muscle cells were fixed with −20°C 100% methanol for 5 min. Alexa Fluor 488 Tyramide SuperBoost kit (Thermo Fisher Scientific, B40912) was also used. In short, endogenous peroxidase activity was first quenched by incubating the fixed muscle cells with 3% hydrogen peroxide for 1 hr, followed by blocking with 2% Bovine Serum Albumin for 1 hr at room temperature. The fixed muscle cells were then stained with MT1-MMP primary antibody at 4°C overnight, followed by incubating with poly-HRP-conjugated secondary antibody for 1 hr at room temperature. Endogenous MT1-MMP signals were developed by incubating the cells with Alexa Fluor 488 tyramide solution for 5 min. Coverslips were then mounted on glass slides with the anti-bleaching reagent fluoromount-G (Thermo Fisher Scientific, 00-4958-02) for later observation.

For mouse diaphragm muscle tissues, the entire diaphragms were dissected and fixed in 2% para-formaldehyde (PFA) in PBS at room temperature for 1 hr, and then incubated with 0.1 M glycine in PBS for 30 min. The fixed tissues were labeled with 0.1 µM tetramethylrhodamine α-bungarotoxin at room temperature for 30 min. After extensive washing with PBS, tissues were labeled with the primary antibody against neurofilament NF200 (1:1000; Sigma, N4142) in the blocking buffer (1% BSA and 0.3% Triton X-100) at 4°C for 2 days. After extensive washing with 0.5% Triton X-100 in PBS, tissues were incubated overnight with Alexa Fluor 488-conjugated secondary antibody at 4°C. Diaphragm muscles were flat-mounted with fluoromount-G onto glass slides.

For rat soleus muscle tissues, the sections were incubated with 0.1 M glycine for 30 min to remove autofluorescence background and permeabilized by methanol. After blocking with 2% BSA and 0.2% Triton X-100 at room temperature for 2 hr, the sections were labeled with primary antibody against MT1-MMP (1:100; Santa Cruz Biotechnology, sc-373908 and 1:100; Millipore, MAB3328), neurofilament NF200 (1:500; Sigma, N4142), and synaptophysin (1:1000; Abcam, ab32127) for 2 days at 4°C. For MT1-MMP pre-incubation, 1 µg recombinant human MT1-MMP (R and D Systems, 918-MP-010) was pre-incubated with 1 µg MT1-MMP antibody for 2 hr at room temperature. After extensive washing with PBS, the muscle sections were labeled with fluorophore-conjugated secondary antibodies (Thermo Fisher Scientific) for 2 hr at room temperature. Muscle sections were mounted with DAPI-Fluoromount-G (Electron Microscopy Sciences, 17984–24).

## Western blot analysis

*Xenopus* embryos at Nieuwkoop and Faber stage 19–22 were homogenized in RIPA buffer in the presence of protease inhibitor cocktail and EDTA, followed by incubation on ice for 5 min. After high-speed centrifugation (15000 x g), the supernatant was collected for protein concentration measurement using a BCA protein assay kit (Thermo Fisher Scientific, 23227). 30 µg protein lysates were used for protein separation by SDS-PAGE, and then blotted onto Immobilon-P membrane (Millipore, IPVH00010). After blocking with 5% non-fat milk in TBST, the blots were probed for the following primary antibodies: MT1-MMP catalytic domain (1:1000; Millipore, AB6005; kindly provided by Professor Timothy Gomez, University of Wisconsin-Madison); CLASP1 (1:250; Abcam, Ab85919); anti-β actin (1:1000; Sigma, A2228); anti-α tubulin (1:2000; Sigma, T6074) overnight at 4°C. After extensive washing with TBST, the blots were reacted with secondary antibodies conjugated with HRP. Signals were detected by ECL western blotting substrate (Thermo Fisher Scientific, 32106), and images were taken by ChemiDoc XRS+ System (Bio-Rad).

## Microscopy

Wide-field fluorescence imaging was performed on inverted microscopes (IX81 or IX83, Olympus) using oil immersion PlanApo 60X NA 1.42 objective lens. On IX81, digital still images were captured by iXon EMCCD camera (Andor) through the software ) through the software cellR (Olympus). On IX83, digital still and time-lapse images were captured by ORCA-Flash4.0 LT+ digital CMOS camera (Hamamatsu) through the software MicroManager (Open Imaging) (*Edelstein et al., 2014*).

In FRAP experiments, total internal reflection fluorescence (TIRF) mode using Axio TIRF unit fitted in an inverted microscope equipped with oil immersion 100X NA 1.46 DIC objective lens was used. Images were captured through Metamorph (Molecular Devices) by Evolve 512 EMCCD camera (Photometrics). To obtain the baseline of fluorescence intensity before photobleaching, 5 to 6 images were taken with 3 s intervals on cells expressing EB1-GFP and 1 s intervals on cells expressing MT1-MMP-pHluorin or MT1-MMP-mCherry. Photobleaching was performed by using Sapphire laser line (488 nm) with 50% laser intensity, which was kept the same for all FRAP experiments. The

photobleaching duration was adjusted based on the signal intensity, ranging from 0.13 to 4.5 s for photobleaching areas between ~ 27 and 34 μm$^2$. After photobleaching, images were taken every 3 s until the fluorescence intensity has reached the plateau level.

In the experimental groups involving photobleaching of aneural AChR clusters, interactive bleaching function was used in a confocal microscope (LSM 800, Carl Zeiss) using 20X NA 1.4 DIC objective lens. Photobleaching was performed by using an argon laser (488 nm) with a 100% laser power. Identical settings were applied in all photobleaching experiments. A fluorescence image was taken immediately after photobleaching to ensure the signals of aneural AChR clusters had been completely bleached.

For imaging of mouse diaphragm muscles and rat soleus muscles, multiple tile-scanned, z-stack images were captured on a confocal microscope (LSM 800, Carl Zeiss) using 20X NA 1.4 DIC objective lens or oil immersion 63X NA 1.4 DIC objective lens. Images were captured with 32-channel GaAsp photomultiplier modules through the software ZEN 2.3 (Carl Zeiss). All acquisition settings (i.e. laser intensity and gain) were kept the same for different experimental groups in the same experiment. All acquired images were processed and analyzed by ImageJ (NIH) or Imaris (Bitplane).

## Quantification and statistical analyses

To quantify the percentage of AChR top and bottom clusters, muscle cells were first identified in the phase-contrast channel, and then the presence of top or bottom aneural AChR clusters was scored in the fluorescence channel.

To quantify the extent of fluorescent gelatin degradation, the intensity of fluorescent gelatin in the region of interest (ROI, created based on the shape and location of AChR clusters) was measured and then normalized with the intensity of fluorescent gelatin at the adjacent area in the same muscle cell. For experiments involving MT1-MMP-mCherry overexpression, the fluorescent gelatin intensity in the region of AChR clusters was normalized with the intensity in the region without cell attachment.

To quantify the change in the area of aneural AChR clusters, ROI was created at the region of AChR clusters in the image taken at the first time point, and then applied to the images taken at subsequent time points for measuring and calculating the percentage change in cluster size.

To quantify the normalized intensity of aneural AChR clusters, the fluorescence intensity of AChRs in the ROI was first measured in all experimental groups and then normalized to the average AChR intensity in control group of the same experiment.

To quantify the density and speed of EB1 comets, ROI for AChR region was created based on the shape and location of AChR clusters, while ROI for non-AChR region was also created at the adjacent area in the same muscle cell. The number and speed of EB1 comets were measured from multiple frames in 7 s time-lapse images using TrackMate plugin in ImageJ (*Tinevez et al., 2017*). The density of EB1 comets was calculated by dividing the number of EB1 comets by the ROI area.

To quantify the FRAP experiments, the relative fluorescence intensity in the photobleached area was first measured and then normalized by the mean fluorescence intensity of the same region in the images acquired before photobleaching. The recovery half-time was determined by the time required for 50% recovery of fluorescence signals in the FRAP regions.

To quantify the number of MT1-MMP-pHluorin surface insertion events, ROI was created at the region of aneural AChR clusters. After photobleaching, the number of MT1-MMP-pHluorin signals in the ROI was measured from multiple frames in 60 s time-lapse images using TrackMate plugin in ImageJ. It was then normalized to the area of aneural AChR clusters.

To quantify the percentage of nerve-muscle contacts with AChR clustering, muscle cells with nerve contact at the bottom surface were first identified in the phase-contrast images, and then the presence of AChR clusters was scored in the fluorescence images.

To quantify the fluorescence intensity of nerve-induced AChR clusters per unit length of nerve-muscle contact, the integrated fluorescence value of AChR signals was first measured, and then it was normalized to the area of nerve-muscle contacts as determined and measured in the phase-contrast images.

To quantify the density of aneural and nerve-induced synaptic AChR clusters in mouse diaphragm muscles, the total number of AChR clusters was first measured from stack images using Imaris software. The number of nerve-induced synaptic AChR clusters was calculated based on the overlapping of signals between AChR clusters and neurofilaments. The number of aneural AChR cluster was then

calculated by subtracting the number of nerve-induced synaptic AChR clusters from the total number of AChR clusters. The calculated numbers were then normalized to the length of the main nerve trunk. Endplate band width was defined by the average distance between two farthest AChR clusters in the muscle fibers along the main nerve trunk. The length of nerve branches was quantified by measuring the average axonal length from the main nerve trunk. Approximately 100 measurements were taken in each diaphragm, and the quantification of the endplate band width and length of branches was performed using the whole diaphragm from at least 3 animals in each genotype.

In all figures, mean and SEM values were shown in the graphs, unless otherwise specified. The numbers of biological replications and the statistical tests applied were specified in the figure legends.

## Acknowledgements

This project was partly supported by the Early Career Grant (27102316) and General Research Fund (17100718 and 17100219) from Research Grants Council of Hong Kong, the Health and Medical Research Fund (04151086) from Food and Health Bureau of Hong Kong, and the Seed Fund Programme for Basic Research from The University of Hong Kong (201601159003, 201711159098, and 201811159078) to CWL. HRK is partly supported by HKU Postgraduate Fellowships in Integrative Medicine. We thank Dr Weichun Lin (UT Southwestern) for sharing the whole-mount diaphragm staining protocol, and Dr Timothy Gomez (University of Wisconsin-Madison) for providing the MT1-MMP antibody.

## Additional information

### Funding

| Funder | Grant reference number | Author |
|---|---|---|
| Research Grants Council, University Grants Committee | 27102316 | Chi Wai Lee |
| Health and Medical Research Fund | 04151086 | Chi Wai Lee |
| Research Grants Council, University Grants Committee | 17100718 | Chi Wai Lee |
| Research Grants Council, University Grants Committee | 17100219 | Chi Wai Lee |

The funders had no role in study design, data collection and interpretation, or the decision to submit the work for publication.

### Author contributions

Zora Chui-Kuen Chan, Data curation, Formal analysis, Validation, Investigation, Visualization, Methodology, Writing - review and editing; Hiu-Lam Rachel Kwan, Data curation, Formal analysis, Validation, Investigation, Visualization, Writing - review and editing; Yin Shun Wong, Formal analysis, Validation, Investigation; Zhixin Jiang, Kin Wai Tam, Resources, Methodology; Zhongjun Zhou, Ying-Shing Chan, Resources; Chi Bun Chan, Resources, Investigation, Methodology; Chi Wai Lee, Conceptualization, Resources, Data curation, Formal analysis, Supervision, Funding acquisition, Validation, Investigation, Visualization, Methodology, Writing - original draft, Project administration, Writing - review and editing

### Author ORCIDs

Zhongjun Zhou (iD) http://orcid.org/0000-0001-7092-8128
Chi Bun Chan (iD) http://orcid.org/0000-0002-2787-9575
Chi Wai Lee (iD) https://orcid.org/0000-0003-4619-3189

## Ethics

Animal experimentation: All experiments involving research animals were carried out in accordance with the approved protocols (#4627-18 and #5149-19) by the Committee on the Use of Live Animals in Teaching and Research (CULATR) of The University of Hong Kong.

## Decision letter and Author response

Decision letter https://doi.org/10.7554/eLife.54379.sa1
Author response https://doi.org/10.7554/eLife.54379.sa2

## Additional files

### Supplementary files

- Transparent reporting form

- Reporting standard 1. Table for statistical tests.

### Data availability

All data generated or analysed during this study are included in the manuscript and supporting files.

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
