## [Decision Letter]

**Acceptance summary:**

Chan et al. have examined the function of actin-enriched podosome-like structures (PLS) and the trafficking of a matrix metalloprotease, MT1-MMP, in regulating AChR organization at the NMJ. Using mostly the *Xenopus* muscle cultures and fluorescence imaging, the authors show that PLS targets MT1-MMP to the membrane surface via the microtubule capturing mechanism involving EB1 and CLASP; subsequently, local ECM remodelling disperses aneural AChR clusters which then act as a source for synaptic AChR clusters. Nerve-muscle co-cultures are also used to explore the effects of interfering with MT1-MMP activity on agrin and nerve-induced AChR cluster formation. Notably, the loss of MT1-MMP activity compromises synaptic AChR cluster formation in vitro and in vivo. Collectively, the findings highlight the crucial role for MT1-MMP in ECM remodeling and AChR re-organization and the underlying cellular and molecular events at the developing NMJ.

**Decision letter after peer review:**

[Editors’ note: the authors submitted for reconsideration following the decision after peer review. What follows is the decision letter after the first round of review.]

Thank you for submitting your work entitled "Podosome-directed MT1-MMP trafficking and surface insertion regulate AChR clustering and remodeling at developing NMJs" for consideration by *eLife*. Your article has been reviewed by three peer reviewers, one of whom is a member of our Board of Reviewing Editors, and the evaluation has been overseen by a Senior Editor. The following individual involved in review of your submission has agreed to reveal their identity: Colin Adrain (Reviewer #3).

Our decision has been reached after consultation between the reviewers. Based on these discussions and the individual reviews below, we regret to inform you that your work will not be considered further for publication in *eLife*, at least in its present form. If you disagree that the further experiments requested here are valuable/necessary, we expect that you will take this work to another journal at this time. If you agree with the general thrust of the reviewers' comments and should decide to do further experimental work, we would be open to receiving a new submission with that work, assuming it is responsive to this critique.

The reviewers have acknowledged the importance of the topic and the extensive amount of work involving expert cell biological analyses. However, all three reviewers have pointed out some shortcomings of the study, especially the lack of compelling experimental evidence for the main conclusion that the AChRs dispersed from aneural clusters serve as the source of synaptic AChR clusters at the NMJ. In addition, the majority of experiments rely on overexpression of MT1-MMP-mCherry, and MT1-MMP is not necessarily selectively targeted to perforations within AChR clusters that raises questions about the model. Finally, mouse experiments using MT1-MMP KO animals are elegant, yet they highlight the possible presynaptic role for MT1-MMP that warrants further clarifications. We hope you will find the reviewer comments appended below to be constructive.

*Reviewer #1:*

Here Chan et al. set out to determine the function of actin-enriched podosome-like structures (PLS) at the NMJ regulating AChR organization. Using mostly the *Xenopus* muscle cultures and fluorescence imaging, the authors perform experiments to conclude that PLS targets MT1-MMP to the membrane surface via the microtubule capturing mechanism involving EB1 and CLASP, which then locally remodels the ECM to disperse aneural AChR clusters. Nerve-muscle cocultures are also used to explore the effects of interfering with MT1-MMP activity on agrin and nerve-induced AChR cluster formation. Collectively, the findings highlight the crucial role for ECM remodeling in AChR organization, and notably, the loss of MT1-MMP activity compromises synaptic AChR cluster formation in vitro and in vivo. Nevertheless, the manuscript is disjointed between characterizations of aneural AChR remodeling and synaptic AChR remodeling in the presence of motoneurons, where most of the effort is spent on the former, and it remains unclear if dispersed aneural AChR serves as a source for synaptic AChRs. Moreover, the logical flow of the proposed events is not necessarily consistent with the data shown, and there are tendencies for over-interpretation of data. Collectively, these issues reduce the overall impact of the present findings.

1) Results, subsection “Topologically Complex Structures of PLS-Associated Aneural AChR Clusters Can Be Induced by Different ECM Proteins”. "These data suggested that different ECM proteins can induce the assembly of PLSs, which in turn regulates the formation and/or remodeling of topologically complex structures of AChR clusters": Only the first half of the statement is supported by the data shown.

2) Figure 2. One should clarify whether block of MMP affects the formation of perforated AChR clusters in any way. Especially with respect to panel G, it is not clear whether the total number of AChR cluster changes over time in control and how this is changed in the presence of MMP inhibitors. In addition to% muscle cells with AChR clusters, one should determine the changes in the number of AChR clusters to demonstrate whether MMP activity is restricted to remodeling of preformed AChR clusters or whether it affects new AChR cluster formation.

3) Results, subsection “MMP-Mediated ECM Degradation Regulates Topological Remodeling of Aneural AChR Clusters”. "These data demonstrated that MMP activity is required for the topological remodeling and dispersal of aneural AChR clusters". This statement is not fully substantiated by the data shown as Figure 2I illustrates that some degree of remodeling does seem to occur even in the presence of BB-94 for most of the experiments.

4) Figure 3A, Figure 4—figure supplement 1 and Figure 5—figure supplement 1. MT1-MMP is not necessarily selectively targeted to perforations within AChR clusters. Also, hotspots of endogenous EB1 are also not necessarily restricted to areas of perforated AChR clusters.

5) Results, subsection “MT1-MMP Activity Precisely Controls the Extent of ECM Degradation and AChR Cluster Formation”. "[…] MT1-MMP-mCherry overexpressing cells showed a higher percentage of top AChR clusters at the expense of ECM-induced bottom AChR clusters […]". The claim that the increase in the proportion of muscle cells that show top AChR clusters occurred "at the expense of bottom AChR clusters" is an overinterpretation of the data. Unless one knows the absolute numbers, one cannot claim for a compensatory effect.

6) Figure 3D. Despite the fact that MMP inhibitors could prevent the reduction in the intensities of gelatin and AChR induced by overexpression of MT1-MMP-mCherry, the finding that muscle cells knocked-down for endogenous MT1-MMP could still support AChR accumulation of AChR at nerve-muscle contacts albeit inefficiently, which presumably occurs via recruiting of aneural AChR clusters (Figure 7) raises concerns about interpretation of MT1-MMP overexpression experiments.

7) Figure 4A-C. The analysis of EB1 dynamics at non-AChR regions should target hotspots of EB1 whose intensity matches those associated to AChR-associated regions (cf. Figure 4—figure supplement 1).

8) Figure 4D and Results, subsection “Cortical Microtubule Capturing Is Mediated by EB1-/CLASP-Dependent Mechanisms at Perforated AChR Clusters”. "[…] EB1-GFP signals were primarily enriched at the edge of perforations of aneural AChR clusters in control cells but at the center of perforations in CLASP-MO muscle cells." No quantified data is presented in support of the claim. Moreover, because AChR clusters are already found to be perforated in CLASP-MO muscle cells, it is not clear how AChR remodeling is dependent on EB1-/CLASP-mediated targeting of MT1-MMP via PLS.

9) Results, subsection “Intracellular Trafficking of MT1-MMP Is Directed to PLS-Associated AChR Clusters”. "[…] these data illustrated that EB1 coordinates the vesicular trafficking of MT1-MMP in cultured muscle cells". The observation of spatiotemporally correlated EB1-GFP and MT1-MMP-mCherry signals alone does not compellingly support the claim here. How prevalent are such behaviors?

10) Results, subsection “Surface Insertion of MT1-MMP Is Correlated with the Formation and Remodeling of AChR Clusters”. "[…] we further determined the causal relationship between surface localization of MT1-MMP and topological remodeling of AChR clusters [...]". The experiments shown here do not establish the causal relationship but rather show whether the two events occur in parallel.

11) Figure 6. As commented above, experiments involving overexpression of MT1-MMP, especially with respect to AChR remodeling, is difficult to interpret due to possible non-specific effects of exogenous overexpression. Note that overexpressed MT1-MMP localization is not limited to AChR cluster regions.

12) Results, subsection “Pharmacological Inhibition of MMP Activity Suppresses the Assembly of Synaptic AChR Clusters and the Disassembly of Aneural AChR Clusters”. "In control cells, we detected a gradual dispersal of aneural AChR clusters upon agrin-bead stimulation for 24h; however, those clusters in agrin bead-contacted muscle cells were greatly stabilized by BB-94 or BB-2516 (Figure 7D and Figure 7—figure supplement 1A)." Contrary to the statement, Figure 7—figure supplement 1A appears to show a lack of difference between control and MMP inhibitors for the decline in the area of aneural AChR clusters.

13) Figure 7—figure supplement 2B and Results, subsection “Pharmacological Inhibition of MMP Activity Suppresses the Assembly of Synaptic AChR Clusters and the Disassembly of Aneural AChR Clusters”. The plot does not seem show a time-dependent increase in aneural AChR clusters in the presence of agrin (4h vs. 8h), which is different from the statement.

*Reviewer #2:*

In this paper the authors tested the idea podosome-like structure (PLS)-directed MT1-MMP trafficking and surface insertion in modulating the assembly and topological remodeling of AChR clusters via focal matrix degradation at developing neuromuscular synapses. First, by culturing frog myotomal tissues on glass coverslips coated with a mixture of ECM proteins, they showed that aneural AChR clusters were formed at the bottom surface of muscle cells in contact with ECL-coated plates (compared with control – coated with poly-D-lysine, PDL) while AChR clutters were sparsely scattered. A majority of ECM-induced AChR clusters appeared perforated; and, ECM markers were enriched in the perforations. By plating muscle cells on FITC-gelatin-coated plates, they showed gelatin degradation in the perforation areas of AChR clusters; and MMP inhibitors, BB-94 and BB-2516, inhibited gelatin degradation and increased muscle fibers with AChR clusters and cluster areas, demonstrating that MMP activity is required for podosome formation and AChR cluster stability. Next, the authors showed that MT1-MMP is enriched in the perforated areas and MT1-MMP mCherry overexpression reduced bottom AChR clusters and increased top clusters. By micro-injecting CLASP-MO and using TIRF, they confirmed CLASP involvement in microtubule capturing at AChR clusters; PLS regulates microtubule capturing and MT1-MMP trafficking to AChR clusters. Then, the authors showed that MMP inhibitors inhibit nerve-induction of AChR clusters and disassembly of aneural clusters. In another set of experiments, MT1-MMP was shown to be necessary for surface expression of new AChR and redistribution of pre-existing AChR. Finally, in MT1-MMP null mice, AChR cluster density and endplate band width were reduced, and nerve terminals were defasciculated. These in vivo data seemed in agreement with in vitro data. Together the authors concluded that MT1-MMP regulates the formation and topological remodeling of postsynaptic specializations at developing NMJs.

This is a cell biology-heavy paper with many cutting edge imaging techniques and in vitro manipulations. Data seemed to be solid. Importantly, some conclusions are supported by data of studies of MT1-MMP mutant mice. However, the following concerns should be addressed before publication.

First, podosome is a morphological phenomenon of some cells in culture. Its in vivo implication remains unclear, in particular at the NMJ. It seemed premature to indicate that in the title. It is misleading to state "podosome-directed MT1-MMP trafficking". I would like to suggest changing to "ECM-directed…".

Second, cell biology data of this paper touched upon a critical issue in NMJ formation – whether the formation of nerve-induced AChR clusters requires the usage of AChR molecules from aneural clutters. The paper showed a correlation for nerve-induction of new clusters versus and disassembly of aneural clusters, which was not new (showed by Peng and colleagues a while back). This paper could be substantially improved by addressing this question. Such in vitro experiments are not too difficult to do.

Third, along the same line, the authors suggested the coupling forming new, nerve-induced clutters with disappearing aneural clusters. However, the paper stopped short in studying MT1-MMP mutant mice. Adding one or more time point(s) should reveal which step(s) is altered. For in vivo studies, imaging and quantification data should be provided regarding "aneural" versus "nerve-induced" clusters. Unfortunately such data are missing in the current version.

Fourth, citation of the papers was generous. Recent reviews on NMJ formation were not cited. The author cited a paper on Yap in tumor. There seemed to be one paper on Yap at NMJ.

Fourth, MT-MMPs have six type (MT1-6). Is MT1-MMP a dominant one? No functional redundancy? Figure 3A, MT1-MMP was distributed everywhere, not specific for AChR clusters. Data for the specificity of the antibody. How about in vivo/muscle staining? At the NMJ? Can some podosome-like phenotypes be observed in muscle cells isolated from MT1-MMP mt mice?

Finally, data such as ECM in AChR podosomes and CLASP involvement were published before. Should this be presented in supplement? Rationales for some experiments such as cortical microtubule in MT1-MMP trafficking could have been better described.

*Reviewer #3:*

This manuscript by Chan and colleagues involves an intricate cell biological dissection of the regulation of acetylcholine receptor (AChR) clustering and remodeling at the neuromuscular junction (NMJ), with a major focus on how this is regulated at the post-synaptic level. This addresses an important topic and in my view, the work is largely well executed and the conclusions robust.

The authors first establish in vitro conditions to characterize the regulation and behavior of AChR clustering under aneural conditions, coming to the conclusion, demonstrated by morpholino and inhibitor studies, that the metalloprotease MT1-MMP (MMP-14) is required for the organization and formation of AChR clusters. The authors further characterize how microtubule capturing mechanisms regulate the dynamics and recruitment of MT1-MMP to the cell surface, as well as the impact on AChR clustering.

Finally, the in vitro/ex vivo analyses are brought together at the end by an examination of the impact of MT1-MMP loss on MNJ formation in mouse embryos. Here the authors conclude that global MT1-MMP embryos have defects in AChR density as well as exhibiting a significant reduced width of end plate bands in developing NMJs. Notably, there are also effects on axonal growth and arborization of the phrenic nerve, making it difficult to conclude, as acknowledged by the authors, that pre-synaptic roles cannot also be ruled out for MT1-MMP.

Overall this is an extensive body of work.

[Editors’ note: further revisions were suggested prior to acceptance, as described below.]

Thank you for submitting your article "Site-directed MT1-MMP trafficking and surface insertion regulate AChR clustering and remodeling at developing NMJs" for consideration by *eLife*. Your article has been reviewed by two peer reviewers, one of whom is a member of our Board of Reviewing Editors, and the evaluation has been overseen by Anna Akhmanova as the Senior Editor. The following individual involved in review of your submission has agreed to reveal their identity: Lin Mei (Reviewer #2).

The reviewers have discussed the reviews with one another and the Reviewing Editor has drafted this decision to help you prepare a revised submission.

Here Chan et al. set out to determine the function of actin-enriched podosome-like structures (PLS) and the trafficking of a matrix metalloprotease, MT1-MMP, in regulating AChR organization at the NMJ. Using mostly the *Xenopus* muscle cultures and fluorescence imaging, the authors demonstrate that PLS targets MT1-MMP to the membrane surface via the microtubule capturing mechanism involving EB1 and CLASP; subsequently, local remodelling of the ECM disperses aneural AChR clusters which then act as a source for synaptic AChR clusters. Nerve-muscle co-cultures are also used to explore the effects of interfering with MT1-MMP activity on agrin and nerve-induced AChR cluster formation. Collectively, the findings highlight the crucial role for ECM remodelling in AChR re-organization, and notably, the loss of MT1-MMP activity compromises synaptic AChR cluster formation in vitro and in vivo.

In the revised manuscript, the authors have made significant efforts to address the previous criticisms by performing new experiments and additional analysis. In particular, the study has been significantly strengthened by the FRAP experiment which supports the idea that synaptic AChRs are recruited from aneural AChRs. Furthermore, the text has been carefully edited to improve the clarity of the data interpretations. Altogether, the study makes a compelling case for a role MT1-MMP in NMJ formation. However, following remaining points require careful consideration.

Essential revisions:

Importantly, "AChR clusters" in the paper are heterogeneous. Therefore, AChR clusters whose formation is triggered by specific stimuli that are found in different locations, may be under different regulatory mechanisms. For example, ECM-induced clusters are located at the bottom surface of myotubes (i.e. surface in direct contact with the ECM), and these clusters seem to be a target of MT1. Whether the effects of MT1 on these "bottom" clusters also apply to AChR clusters on the top surface and/or nerve-induced clusters (which usually are located on the lateral sides of myotubes) remain a critical issue. While it may be beyond the scope of the current study to experimentally clarify the role of MT1 on different AChR cluster types, at the very least, the authors should define these types of AChR clusters in the Introduction. A full discussion of whether and/how MT1 regulates all or just one of the AChR cluster types should be included along with a cartoon.

---

## [Author Response]

[Editors’ note: the authors resubmitted a revised version of the paper for consideration. What follows is the authors’ response to the first round of review.]

Four major sets of new data are included in this revision in response to reviewers’ comments:

1) Contribution of AChRs from aneural to synaptic clusters. All reviewers raised an important question regarding whether aneural AChR clusters serve as a source for the assembly of synaptic AChR clusters. We agree that our previous manuscript lacks the compelling experimental evidence to show the direct contribution of aneural AChR clusters at developing NMJs. In this revision, we have performed this key experiment using laser-based photobleaching approach to demonstrate the differential contribution of AChR molecules from aneural clusters and diffuse AChRs for the assembly of nerve-induced synaptic AChR clusters. Importantly, we have also investigated the involvement of MT1MMP in this recruitment process. The new data are now presented in Figure 9.

2) Aneural versus synaptic AChR clusters in MT1-MMP^-/-^ knockout study. We agree with reviewer #2 that our manuscript could be further strengthened by providing additional imaging and quantification data to show both aneural and synaptic AChR clusters at developing NMJs in vivo. As such, we have now performed new quantitative analyses on our existing and new imaging data to measure the numbers of aneural and synaptic AChR clusters between wild-type and MT1-MMP knockout animals. The new in vivo analyses are now presented in Figure 10C-G.

3) Possible presynaptic role of MT1-MMP at NMJs in vivo. We appreciate the comments from reviewer #3 indicating that it is outside the scope of our present manuscript to determine whether MT1-MMP plays a presynaptic role at developing NMJs in vivo. As suggested by the reviewer, we have now performed the immunostaining experiment using rat leg muscles. Our new results indicated that denervation caused only a slight reduction of MT1-MMP signals at the NMJs, suggesting MT1-MMP is expressed in postsynaptic muscles primarily, but also in presynaptic motor neurons at a lower level. The new in vivo data are now presented in Figure 10B.

4) Spatial localization of endogenous and overexpressed MT1-MMP in AChRclusters. Reviewers #1 and #2 raised concerns about the spatial localization of endogenous and overexpressed MT1-MMP, as their signals are not selectively restricted to the perforations of aneural AChR clusters. In the revised manuscript, we have now provided additional data using MT1-MMP antibody that was pre-incubated with recombinant MT1-MMP proteins to demonstrate the specificity of MT1-MMP signals at NMJs in vivo in Figure 10A. Moreover, we have further clarified that signals of MT1-MMP immunostaining and MT1-MMP-mCherry overexpression are derived from both surface and intracellular MT1-MMP proteins, explaining why it is reasonable that endogenous and overexpressed MT1-MMP signals are not restricted to AChR clusters.

Reviewer #1:[…] The manuscript is disjointed between characterizations of aneural AChR remodeling and synaptic AChR remodeling in the presence of motoneurons, where most of the effort is spent on the former, and it remains unclear if dispersed aneural AChR serves as a source for synaptic AChRs.

We thank the reviewer for the comment. In this revision, we have further provided a direct evidence demonstrating the contribution of AChRs from the aneural to synaptic clusters at developing NMJs in Figure 9. Our new results showed that a majority of AChRs at the nerve-muscle contacts are recruited from aneural AChR clusters, and this recruitment process is significantly reduced in MT1-MMP knockdown muscle cells.

Moreover, the logical flow of the proposed events is not necessarily consistent with the data shown, and there are tendencies for over-interpretation of data. Collectively, these issues reduce the overall impact of the present findings.

We have now modified the schematic diagram (Figure 11B) by including the relevant data presented in our main figures that support each of the individual proposed events in this logical flow diagram. Moreover, we have toned down several overstatements throughout the revised manuscript. With these changes, we hope that the reviewer and the general audience would better appreciate the overall impact of this study.

1) Results, subsection “Topologically Complex Structures of PLS-Associated Aneural AChR Clusters Can Be Induced by Different ECM Proteins”. "These data suggested that different ECM proteins can induce the assembly of PLSs, which in turn regulates the formation and/or remodeling of topologically complex structures of AChR clusters": Only the first half of the statement is supported by the data shown.

We apologize for the over-interpretation in this sentence. In this study, we have demonstrated that the formation of bottom AChR clusters could be effectively induced by all ECM proteins tested (Figure 1B), and we have identified PLS localization in a majority of perforated AChR clusters induced by different ECM proteins (Figure 1D-E). Therefore, we have revised the sentence - “These data suggest that different ECM proteins can induce the assembly of PLSs, which may in turn regulate the formation of topologically complex structures of AChR clusters”.

2) Figure 2. One should clarify whether block of MMP affects the formation of perforated AChR clusters in any way. Especially with respect to panel G, it is not clear whether the total number of AChR cluster changes over time in control and how this is changed in the presence of MMP inhibitors. In addition to % muscle cells with AChR clusters, one should determine the changes in the number of AChR clusters to demonstrate whether MMP activity is restricted to remodeling of preformed AChR clusters or whether it affects new AChR cluster formation.

We agree with the reviewer that our original Figure 2G did not clearly show the total number of AChR clusters changes over time in control and in the presence of MMP inhibitors. In this revision, we have modified the presentation of the data in the original Figure 2G. Two separate panels are now added to show the effects of MMP inhibition on the formation (Figure 2G) and remodeling (Figure 2H) of AChR clusters. Moreover, we have included the new quantitative data as Author response image 1, showing that there is no significant difference in the total number of AChR clusters between control and MMP inhibitor-treated muscle cells. These data indicated that MMP inhibition affects the remodeling of preformed AChR clusters, rather than the formation of new AChR clusters.

**Author response image 1. respfig1:** Quantitative analysis on the total AChR clusters in cultured cells treated with MMP inhibitors. By monitoring muscle cells over 4 days in culture, AChR clusters located in the top versus bottom surface of the muscle cells were quantified in different experimental groups. Data are represented as mean ± SEM. n = 150 muscle cells in each experimental group from 3 independent experiments.

3) Results, subsection “MMP-Mediated ECM Degradation Regulates Topological Remodeling of Aneural AChR Clusters”. "These data demonstrated that MMP activity is required for the topological remodeling and dispersal of aneural AChR clusters". This statement is not fully substantiated by the data shown as Figure 2I illustrates that some degree of remodeling does seem to occur even in the presence of BB-94 for most of the experiments.

It is true that a small degree of remodeling of aneural AChR clusters can be detected in the presence of MMP inhibitors. This is due to the fact that the activity of MMPs cannot be completely blocked by pharmacological inhibitors. However, it should be noted that we observed a significant inhibition of AChR cluster remodeling in response to BB-94 treatment (Figure 2I-J). Therefore, we have toned down the quoted sentence – “These data demonstrated that MMP activity is involved in the topological remodeling and dispersal of aneural AChR clusters”.

4) Figure 3A, Figure 4—figure supplement 1 and Figure 5—figure supplement 1. MT1-MMP is not necessarily selectively targeted to perforations within AChR clusters. Also, hotspots of endogenous EB1 are also not necessarily restricted to areas of perforated AChR clusters.

We thank the reviewer for this comment. We would like to emphasize that MT1-MMP immunostaining data in Figure 3A and Figure 5—figure supplement 1 indicated the signals from not only surface MT1-MMP, but also intracellular MT1-MMP at the vesicular compartments. Therefore, it was expected that MT1-MMP signals in Figure 3A are not selectively restricted to the perforations of aneural AChR clusters. We have emphasized this in the third paragraph of the Discussion.

Regarding the endogenous EB1 signals, we also expected that EB1 signals are not selectively restricted to perforated AChR clusters. As EB1 is a microtubule plus-end-binding protein responsible for regulating microtubule dynamics, endogenous EB1 comets could be found throughout the entire cell. In addition to the location of aneural AChR clusters, it is known that focal adhesions at the cell periphery can also regulate cortical microtubule capturing, representing another possible site of EB1 hotspots in muscle cells. We have now clarified this in the fourth paragraph of the Discussion.

5) Results, subsection “MT1-MMP Activity Precisely Controls the Extent of ECM Degradation and AChR Cluster Formation”. "[…] MT1-MMP-mCherry overexpressing cells showed a higher percentage of top AChR clusters at the expense of ECM-induced bottom AChR clusters […]". The claim that the increase in the proportion of muscle cells that show top AChR clusters occurred "at the expense of bottom AChR clusters" is an overinterpretation of the data. Unless one knows the absolute numbers, one cannot claim for a compensatory effect.

We agree with the reviewer on this. This statement has been toned down – “MT1-MMP-mCherry overexpressing cells showed a lower percentage of ECM-induced bottom AChR clusters, but a higher percentage of top AChR clusters, in comparison to the wild-type control cells”.

6) Figure 3D. Despite the fact that MMP inhibitors could prevent the reduction in the intensities of gelatin and AChR induced by overexpression of MT1-MMP-mCherry, the finding that muscle cells knocked-down for endogenous MT1-MMP could still support AChR accumulation of AChR at nerve-muscle contacts albeit inefficiently, which presumably occurs via recruiting of aneural AChR clusters (Figure 7) raises concerns about interpretation of MT1-MMP overexpression experiments.

We thank the reviewer for pointing this out. We would like to clarify that MT1-MMP serves different roles in regulating the aneural (Figure 3) and synaptic (Figures 7-8) AChR clusters. As the reviewer pointed out, nerve-muscle contacts in the chimeric co-cultures of MT1-MMP-MO muscles and wild-type neurons could still allow the accumulation of some AChRs, albeit inefficiently. This could be explained by: (1) the partial knockdown of endogenous muscle MT1-MMP level by MT1-MMP MO (Figure 8—figure supplement 1); (ii) the recruitment of diffuse AChRs to nerve-muscle contacts, although it is a minor contribution, via MT1-MMP-independent processes (Figure 9). Therefore, it is not contradictory with the findings of MT1-MMP overexpression in aneural AChR cluster formation (Figure 3). We have now clarified this in the subsection “Postsynaptic MT1-MMP is a regulator of NMJ development”.

7) Figure 4A-C. The analysis of EB1 dynamics at non-AChR regions should target hotspots of EB1 whose intensity matches those associated to AChR-associated regions (cf. Figure 4—figure supplement 1).

Apart from the region of aneural AChR clusters, focal adhesion structures located at the cell periphery are the other site of EB1 hotspots in the muscle cells. Therefore, we performed the suggested analysis to compare EB1-GFP dynamics between AChR and focal adhesion regions. As shown in Author response image 2, the speed of EB1 comets was comparable between AChR and focal adhesion regions. However, CLASP knockdown significantly increased the speed of EB1 comets in AChR regions but not in focal adhesion regions. These data suggested that while cortical microtubule capturing is detected at both aneural AChR clusters and focal adhesion structures, the former is more sensitive to CLASP expression level.

**Author response image 2. respfig2:** Cortical microtubule capturing at AChR and focal adhesion regions. (**A**) Representative TIRF images showing the hotspots of EB1-GFP comets in AChR clusters and focal adhesion (FA) structures in control or CLASP-MO muscle cells. (**B**) Quantitative analyses showing the speed of EB1-GFP comets in AChR cluster and focal adhesion regions between control and CLASP-MO muscle cells. n = 15 (Control) and 13 (CLASP-MO) cells from 4 independent experiments. Scale bar represents 10 μm. Data are represented as mean ± SEM. Student’s t-test, * represents p ≤ 0.05.

8) Figure 4D and Results, subsection “Cortical Microtubule Capturing Is Mediated by EB1-/CLASP-Dependent Mechanisms at Perforated AChR Clusters”. "[…] EB1-GFP signals were primarily enriched at the edge of perforations of aneural AChR clusters in control cells but at the center of perforations in CLASP-MO muscle cells." No quantified data is presented in support of the claim.

To better show the spatial patterns of EB1-GFP signals at the perforations of aneural AChR clusters in control versus CLASP-MO muscle cells, we have now plotted the fluorescence intensity profiles of EB1-GFP and AChR across a perforated region of AChR clusters. Our new analysis clearly indicated the enrichment of EB1-GFP at the edge of perforations in control cells, but at relatively closer to the centre of perforations in CLASP-MO muscle cells. The new data are now presented in Figure 4E.

However, different perforations in different, and even the same, aneural AChR clusters exhibit a great diversity and variations in terms of their size, shape, and intensity difference. This prevented us from performing other quantitative analyses to further support our claim.

Moreover, because AChR clusters are already found to be perforated in CLASP-MO muscle cells, it is not clear how AChR remodeling is dependent on EB1-/CLASP-mediated targeting of MT1-MMP via PLS.

Western blotting analysis in Figure 4—figure supplement 2 showed that CLASP protein level is largely reduced, but not completed abolished, in the condition of morpholino-mediated CLASP knockdown. The residual CLASP protein level explained why perforated AChR clusters were able to form in CLASP-MO muscle cells. Since CLASP can control other cellular events (e.g. to maintain a stable spindle position in mitosis (Samora et al., 2011)), it is not desirable to further increase the knockdown efficiency of endogenous CLASP in our experiments.

9) Results, subsection “Intracellular Trafficking of MT1-MMP Is Directed to PLS-Associated AChR Clusters”. "[…] these data illustrated that EB1 coordinates the vesicular trafficking of MT1-MMP in cultured muscle cells". The observation of spatiotemporally correlated EB1-GFP and MT1-MMP-mCherry signals alone does not compellingly support the claim here. How prevalent are such behaviors?

As stated in our response to comment #4 above, EB1 is a microtubule plus-end binding protein responsible for regulating microtubule dynamics. The intracellular trafficking of MT1-MMP vesicles can be mediated throughout the entire microtubule networks. Even though we were able to observe cases when EB1 comets search and capture MT1-MMP vesicles (Figure 5A), the exact location of MT1-MMP vesicles is not selectively restricted at/near the plus-end of microtubules. Therefore, it is not feasible to perform quantitative analysis on the prevalence of such behaviors.

10) Results, subsection “Surface Insertion of MT1-MMP Is Correlated with the Formation and Remodeling of AChR Clusters”. "[…] we further determined the causal relationship between surface localization of MT1-MMP and topological remodeling of AChR clusters [...]". The experiments shown here do not establish the causal relationship but rather show whether the two events occur in parallel.

We apologize for this overstatement. The quoted sentence has been toned down – “[…] we further determined the correlation between surface localization of MT1-MMP and topological remodeling of AChR clusters […]”.

11) Figure 6. As commented above, experiments involving overexpression of MT1-MMP, especially with respect to AChR remodeling, is difficult to interpret due to possible non-specific effects of exogenous overexpression. Note that overexpressed MT1-MMP localization is not limited to AChR cluster regions.

We understand the concern about the possible non-specific effects of MT1-MMP overexpression for the study of AChR cluster remodeling. In contrast to MT1-MMP-mCherry overexpression that showed both surface and intracellular MT1-MMP signals (Figure 5), the localization of MT1-MMP-pHluorin signals at a low expression level was indeed more restricted to AChR cluster regions (Figure 6A). Therefore, to minimize the non-specific effects of exogenous overexpression, only those cells with low MT1-MMP-pHluorin expression levels were chosen for the study of AChR cluster remodeling (Figure 6D-H). Taking all these into account, MT1-MMP-pHluorin is still considered as a very useful tool to demonstrate the surface insertion of MT1-MMP in our experiments. We have now further emphasized this point in the Discussion section.

12) Results, subsection “Pharmacological Inhibition of MMP Activity Suppresses the Assembly of Synaptic AChR Clusters and the Disassembly of Aneural AChR Clusters”. "In control cells, we detected a gradual dispersal of aneural AChR clusters upon agrin-bead stimulation for 24h; however, those clusters in agrin bead-contacted muscle cells were greatly stabilized by BB-94 or BB-2516 (Figure 7D and Figure 7—figure supplement 1A)." Contrary to the statement, Figure 7—figure supplement 1A appears to show a lack of difference between control and MMP inhibitors for the decline in the area of aneural AChR clusters.

We realize that plotting the three experimental groups as a mixed population in the original figure was difficult for data interpretation. We have now presented the data points in each experimental groups side by side in Figure 7—figure supplement 1A. Additionally, statistical analyses have been performed to show that BB-94 and BB-2516 significantly suppressed the decline in the area of aneural AChR clusters upon agrin bead stimulation, compared with the control group.

13) Figure 7—figure supplement 2B and Results, subsection “Pharmacological Inhibition of MMP Activity Suppresses the Assembly of Synaptic AChR Clusters and the Disassembly of Aneural AChR Clusters”. The plot does not seem show a time-dependent increase in aneural AChR clusters in the presence of agrin (4h vs. 8h), which is different from the statement.

We have now clarified this confusing statement in the revised manuscript – “[…] we performed immunostaining experiments, in which we demonstrated an increase of endogenous MT1-MMP signals at aneural AChR clusters upon agrin stimulation (Figure 7—figure supplement 2)”.

Reviewer #2:[…] This is a cell biology-heavy paper with many cutting edge imaging techniques and in vitro manipulations. Data seemed to be solid. Importantly, some conclusions are supported by data of studies of MT1-MMP mutant mice. However, the following concerns should be addressed before publication.First, podosome is a morphological phenomenon of some cells in culture. Its in vivo implication remains unclear, in particular at the NMJ. It seemed premature to indicate that in the title. It is misleading to state "podosome-directed MT1-MMP trafficking". I would like to suggest changing to "ECM-directed…".

We agree with the reviewer that it is premature to use “podosome-directed MT1-MMP trafficking” in the title. While we have considered the suggestion of using “ECM-directed MT1-MMP trafficking”, our data demonstrated that MT1-MMP is indeed targeted to and inserted at the perforated regions of AChR clusters, where extensive degradation of ECM proteins is spatially detected. Therefore, using “ECM-directed MT-MMP trafficking…” does not truly reflect our main findings in this study. In this revised manuscript, we have decided to revise the title as, “Site-directed MT1-MMP trafficking and surface insertion regulate AChR clustering and remodeling at developing NMJs”.

Second, cell biology data of this paper touched upon a critical issue in NMJ formation – whether the formation of nerve-induced AChR clusters requires the usage of AChR molecules from aneural clutters. The paper showed a correlation for nerve-induction of new clusters versus and disassembly of aneural clusters, which was not new (showed by Peng and colleagues a while back). This paper could be substantially improved by addressing this question. Such in vitro experiments are not too difficult to do.

We thank the reviewer for this excellent suggestion. In this revision, we have performed this key experiment using photobleaching approach to demonstrate the differential contribution of AChR molecules from aneural clusters and diffuse AChRs for the assembly of nerve-induced synaptic AChR clusters. The new data are now presented in Figure 9.

Third, along the same line, the authors suggested the coupling forming new, nerve-induced clutters with disappearing aneural clusters. However, the paper stopped short in studying MT1-MMP mutant mice. Adding one or more time point(s) should reveal which step(s) is altered. For in vivo studies, imaging and quantification data should be provided regarding "aneural" versus "nerve-induced" clusters. Unfortunately such data are missing in the current version.

We agree with the reviewer that imaging and quantifying aneural versus synaptic AChR clusters at different embryonic stages would be of help to further support the coupling events between aneural AChR cluster dispersal and synaptic AChR cluster formation at developing NMJs. Thus, we have now classified all AChR signals into aneural and synaptic AChR clusters in wild-type and MT1-MMP^-/-^ diaphragm muscles at E13.5 and E18.5, which are now presented in Figure 10C-E. At E13.5, we observed a significant increase in the density of aneural AChR clusters in MT1-MMP^-/-^ compared with that in the wild-type animals (Figure 10D). In contrast, the density of synaptic AChR clusters in MT1-MMP^-/-^ was significantly lower than that in wild-type animals at E18.5 (Figure 10E). This new quantitative data further supported that MT1-MMP is involved in the recruitment of aneural AChR clusters to nerve-induced synaptic AChR clusters at developing NMJs in vivo.

In addition, we have tried to add an additional time point (E15.5) for examining the AChR clusters in embryonic diaphragm muscles between wild-type and MT1-MMP-deficient mice. Unfortunately, because of the low breeding performance of our existing MT1-MMP^-/-^ colonies, we were able to obtain only one homozygous MT1-MMP knockout embryo in the past 6 months. From our preliminary data as shown in Author response image 3, the effects on the density of aneural versus synaptic AChR clusters between the wild-type and MT1-MMP^-/-^ mice at E15.5 showed a similar trend as what we observed using E13.5 mice (Figure 10D-E). With the new analysis suggested by the reviewer as stated above, we think that the comparison between E13.5 and E18.5 is sufficient to demonstrate the coupling events between the dispersal of aneural AChR clusters and the formation of synaptic AChR clusters.

Nevertheless, if the reviewers consider this as an essential experiment for the revision, we are in the process of replacing the poor breeders, and therefore we shall be able to obtain more homozygous MT1-MMP knockout mice for this study in the next couple of months.

**Author response image 3. respfig3:** Aneural versus synaptic AChR clusters in MT1-MMP^-/-^ diaphragm muscles at E15.5. (**A**) Representative confocal images showing the AChR pre-patterns (red) and synaptic AChR clusters (yellow) in whole-mount diaphragms from control and MT1-MMP^-/-^ mice at E15.5. Whole-mount tissues were stained for AChR and neurofilament (NF). (B-E) Quantification on the density of aneural (**B**) versus synaptic (**C**) AChR clusters, the width of end-plate bands (**D**), and the length of axonal branches (**E**) in diaphragm muscles between wild-type and MT1-MMP^-/-^ mouse embryos. n = 3 (WT) and 1 (MT1-MMP^-/-^) embryos from 1 experiment only. Data are represented as mean ± SEM (WT). Scale bar represents 50 μm.

Fourth, citation of the papers was generous. Recent reviews on NMJ formation were not cited. The author cited a paper on Yap in tumor. There seemed to be one paper on Yap at NMJ.

We apologize for missing out some important, recently published review and research papers on NMJ formation. We have now included several recent reviews on NMJ development (Li et al., 2018; Wu et al., 2010), and the research article on Yap at NMJ (Zhao et al., 2017) in the revised manuscript.

Fifth, MT-MMPs have six type (MT1-6). Is MT1-MMP a dominant one? No functional redundancy? Figure 3A, MT1-MMP was distributed everywhere, not specific for AChR clusters. Data for the specificity of the antibody. How about in vivo/muscle staining? At the NMJ?

Among the six membrane-type MMPs, MT1-MMP has been widely studied since its identification and is involved in ECM remodeling in physiological and pathological processes. In the present study, morpholino-mediated knockdown of MT1-MMP alone was able to significantly inhibit the formation and dispersal of synaptic and aneural AChR clusters, respectively. Besides, MT1-MMP knockout mice develop multiple abnormalities and die between 50-90 postnatal days (Holmbeck et al., 1999), suggesting that the possibility of functional redundancy by other MMP isoforms is low. We have now further clarified this point in the sixth paragraph of the Discussion.

Regarding the specificity of MT1-MMP antibodies, we have now provided additional data using MT1-MMP antibody that was pre-incubated with recombinant MT1-MMP protein to demonstrate the specificity of MT1-MMP signals at NMJs in vivo in Figure 10A.

Can some podosome-like phenotypes be observed in muscle cells isolated from MT1-MMP mt mice?

We have now performed immunostaining experiments to examine the localization of ADF/cofilin and talin, core and cortex markers of podosome-like structures respectively, at AChR clusters in hindlimb muscles of wild-type and MT1-MMP^-/-^ postnatal mice. Consistent with a previous study using adenovirus-expressing actin-GFP in tibialis anterior muscles in postnatal day 8 mice (Proszynski et al., 2009), the spatial localization of synaptic podosomes at the NMJs in vivo is not as obvious as that in cultured muscles. Nevertheless, we were able to detect elevated signals of ADF/cofilin and talin associated with synaptic AChR clusters in hindlimb muscles from both wild-type and MT1-MMP^-/-^ mice at postnatal day 11. This indicated that MT1-MMP deficiency causes no or minimal change in the presence of podosome-like structures at NMJs during early postnatal development.

**Author response image 4. respfig4:** Localization of PLS core and cortex markers at AChR clusters in postnatal mice. Representative confocal images showing the enrichment of PLS core marker, ADF/cofilin (top panels), and cortex marker, talin (bottom panels), at synaptic AChR clusters in longitudinal cryosections of hindlimb muscles isolated from P11 wild-type or MT1-MMP^-/-^ mice. Maximum projection intensity images of 12 z-stack confocal frames with 0.48 µm intervals were constructed using ZEN 2.3 (Carl Zeiss). Scale bar represents 10 μm.

Finally, data such as ECM in AChR podosomes and CLASP involvement were published before. Presented in supplement? Rationales for some experiments such as cortical microtubule in MT1-MMP trafficking could have been better described.

While we acknowledge the previously published works on the involvement of ECM in podosome assembly at AChR clusters and CLASP involvement in this field, we believe that our new data presented in this study provided some novel insights into these processes.

Firstly, our data showed that perforated AChR clusters can be effectively induced by gelatin (Figure 1B and C), an ECM protein that was previously shown to induce simple AChR plaques only (Kummer et al., 2004). Secondly, our TIRF-FRAP experiments demonstrated that EB1-GFP signals were primarily enriched at the edge of perforations of aneural AChR clusters in control cells, but at the center of perforations in CLASP-MO muscle cells (Figure 4D-E). As such, we would prefer to keep these data in the main figure.

We have now further elaborated the rationales of experiments investigating cortical microtubules in MT1-MMP trafficking as suggested – “As microtubules serve as major tracks for vesicular trafficking in mammalian cells, the cortical microtubule organization plays a crucial role in the targeted delivery of secretory and membrane proteins via exocytosis”.

[Editors’ note: what follows is the authors’ response to the second round of review.]

Essential revisions:Importantly, "AChR clusters" in the paper are heterogeneous. Therefore, AChR clusters whose formation is triggered by specific stimuli that are found in different locations, may be under different regulatory mechanisms. For example, ECM-induced clusters are located at the bottom surface of myotubes (i.e. surface in direct contact with the ECM), and these clusters seem to be a target of MT1. Whether the effects of MT1 on these "bottom" clusters also apply to AChR clusters on the top surface and/or nerve-induced clusters (which usually are located on the lateral sides of myotubes) remain a critical issue. While it may be beyond the scope of the current study to experimentally clarify the role of MT1 on different AChR cluster types, at the very least, the authors should define these types of AChR clusters in the Introduction. A full discussion of whether and/how MT1 regulates all or just one of the AChR cluster types should be included along with a cartoon.

We thank the Reviewing Editor and reviewers for raising this crucial point and providing an opportunity for us to clarify this. In most of our experiments, aneural AChR clusters located at the basal muscle surface in direct contact with ECM proteins (i.e. bottom AChR clusters), as well as synaptic AChR clusters located at either the basal surface or the lateral side of muscle cells, were investigated. The central idea of this study is to test whether ECM degradation mediated by MT1-MMP regulates AChR cluster formation and remodeling.

Although it is true that a minority of aneural AChR clusters can be found at the top muscle surface, where no ECM proteins are directly involved, we consider that top AChR clusters are spontaneously formed with no physiological relevance. Thus, we did not investigate the role of MT1-MMP in the formation and remodeling of ECM-independent, top AChR clusters.

As for nerve-induced AChR clusters, most nerve contacts in *Xenopus* primary cultures are located at either the basal surface or the lateral side of muscle cells, in agreement with previous studies (Anderson and Cohen, 1977; Cohen et al., 1987). Since it is uncommon to find nerve contacts at the top surface of cultured muscle cells, they were not included in our study.

We have now added several sentences in different parts of our revised manuscript to further clarify these points:

Introduction:

1) “These aneural AChR clusters, located at the bottom surface of cultured muscles in direct contact with ECM proteins, …”.

2) “It is worth to note that a small proportion of aneural AChR clusters can also be identified at the top surface of cultured muscles, and these spontaneously formed clusters are likely mediated through ECM- and PLS-independent mechanisms”.

Discussion:

1) “In agreement with previous studies (Anderson and Cohen, 1977; Cohen et al., 1987), most nerve contacts in *Xenopus* primary cultures are located at either the basal surface or the lateral side of muscle cells, which are influenced by MMP-mediated proteolytic degradation of ECM proteins”.

2) “In a minority of cases, some aneural and nerve-induced AChR clusters can be found at the top muscle surface, where no ECM proteins are directly involved. Particularly, those top aneural AChR clusters are spontaneously formed with possibly no physiological relevance, whether MT1-MMP regulates the formation and remodeling of ECM-independent top AChR clusters remains unknown”.

Schematic cartoon:

1) The schematic diagram has been significantly revised to indicate the role of MT1-MMP in the recruitment of AChRs from bottom aneural clusters to bottom synaptic clusters via focal ECM degradation. This diagram is now presented in Figure 11A, as this does not belong to the supplement of a particular figure.